

# Long-term (2010-2021) lidar observations of stratospheric aerosols at Wuhan, China

Yun He[1,2,3,a,*], Dongzhe Jing[1,2,3,a], Zhenping Yin[4], Kevin Ohneiser[5], Fan Yi[1,2,3]

[1]School of Electronic Information, Wuhan University, Wuhan 430072, China
[2]Key Laboratory of Geospace Environment and Geodesy, Ministry of Education, Wuhan 430072, China.
[3]State Observatory for Atmospheric Remote Sensing, Wuhan 430072, China.
[4]School of Remote Sensing and Information Engineering, Wuhan University, Wuhan 430072, China.
[5]Leibniz Institute for Tropospheric Research, Permoserstr. 15, Leipzig 04318, Germany
[a]These authors contributed equally to this work.

*Correspondence to:* Yun He (heyun@whu.edu.cn)

**Abstract.** Stratospheric aerosols are long-lived and play a critical role in the global radiation budget. Over the past decade, contributions to stratospheric aerosols from different sources have changed due to weaker volcanic activity and more frequent wildfire events. However, long-term observations of stratospheric aerosols and monitoring of major emission events remain insufficient, particularly at middle and low latitudes. In this study, we analyze the vertical distribution, optical properties, and

radiative forcing of stratospheric aerosols using observations from a ground-based polarization lidar in Wuhan (30.5°N, 114.4°E) from 2010 to 2021.

The stratospheric aerosol optical depth (sAOD) generally stabilized around 0.0023 without significant annual variation. Several cases of volcanic aerosol and wildfire-induced smoke were observed. Volcanic aerosols from the Nabro (2011) and Raikoke (2019) eruptions (both in boreal summer) increased the sAOD to 4.8 times the background level during the

stratospheric-quiescent period (January 2013 to August 2017). Tracers of smoke from the Canadian wildfire in the summer of 2017 was observed twice: at 19-21 km on 14-17 September and at 20-23 km on 28-31 October, with plume-isolated AOD of 0.002-0.010 and particle linear depolarization ratio $\delta_p$ of 0.14-0.18, indicating the dominance of non-aged smoke particles. During these summertime events, the injected stratospheric aerosols were captured by the large-scale Asian monsoon anticyclone (AMA), confining the transport pathway to mid-latitude Asia. On 8-9 November 2020, smoke plumes originating

from the California wildfire in October 2020 appeared at 16-17 km, with a plume-isolated AOD of 0.007 and a mean $\delta_p$ of 0.13. Regarding seasonal variation, the sAOD in the cold half-year (0.0026) is 24% larger than in the warm half-year (0.0021) due to stronger meridional transport of stratospheric aerosols from the tropics to middle latitudes. The stratospheric radiative forcing was -0.05 W·m$^{-2}$ during the stratospheric-quiescent period and increased to -0.28 W·m$^{-2}$ when volcanic aerosols were largely injected. These findings contribute to our understanding of the sources and transport patterns of stratospheric aerosols

over mid-latitude Asia and serve as important database for the validation of model outputs.



## 1. Introduction

The stratospheric aerosol layer (SAL) extends from the tropopause up to approximately 30-km height and is long-lasting, with a residence time of several months to years (Junge, 1960; Junge and Manson, 1961; Hitchman et al., 1994; Kremser et al., 2016). Stratospheric aerosols play a critical role in the global radiation budget by scattering incoming solar radiation back to space, resulting in the cooling of the near-surface and lower atmosphere (Thompson and Solomon, 2009; Solomon et al., 2011). In addition, stratospheric aerosols can activate heterogeneous chemistry by serving as a reaction surface, leading to stratospheric ozone depletion (Hofmann and Solomon, 1989; Tritscher et al., 2021; Ohneiser et al., 2022).

The SAL mainly consists of sulfate aerosols, which are formed from $SO_2$ and ash emitted by volcanic eruptions via oxidation and condensation (Gorkavyi et al., 2021). Moreover, other sources also contribute to the SAL, including smoke particles emitted from wildfires, carbonyl sulfide (OCS) and dimethyl sulfide (DMS) from the sea, $SO_2$ from anthropogenic activities, emissions from air traffic, and dust aerosols from Asia and Africa (SPARC, 2006; Peterson et al., 2018; Trickl et al., 2024). In general, the long-term characteristics of stratospheric aerosols intermittently show the stratospheric background level (i.e., during the stratospheric-quiescent period) and the SAL perturbations caused by significant volcanic eruptions. Therefore, it is of great importance to evaluate the background level of stratospheric aerosols by taking advantage of the occasional stratospheric-quiescent periods.

Deshler et al. (2006) found no significant change in background stratospheric aerosol levels from the 1970s to 2004. Similarly, Khaykin et al. (2017) reported that the stratospheric aerosol optical depth (sAOD) in France during 1997-2003 was $2.37 \times 10^{-3}$, the lowermost since 1970, which can be considered a reference for background levels. In contrast, stratospheric aerosols showed an increasing trend in the first decade of the 21st century due to several intense volcanic eruptions (Hofmann et al., 2009; Solomon et al., 2011; Vernier et al., 2011). Another volcanic-quiescent period was 2013-2019, between the eruptions of Nabro in 2011 and Raikoke in 2019. Meanwhile, intense wildfire events became more frequent, such as the Canadian wildfire in 2017, the Siberian wildfire in 2019 the Australian wildfire in 2019, and the California wildfire in 2020 (Ansmann et al., 2022; Mamouri et al, 2023; Ohneiser et al., 2020, 2022), which injected a mass of smoke particles into the stratosphere via pyro-cumulonimbus clouds (Fromm et al., 2003, 2010). Smoke particles can increase the particle number and surface area concentration of polar stratospheric clouds (PSCs), resulting in ozone depletion by halogen activation reactions on the surface of liquid PSC particles (Ansmann et al., 2022). Additionally, previous studies have speculated that the increasing Asian $SO_2$ emissions may also contribute to the stratospheric aerosol levels (Vernier et al., 2015), which should be further examined with observations (Kremser et al., 2016). In consequence, continuous observations in the second decade of 21st century provide a valuable opportunity to estimate the contributors to stratospheric aerosol levels aside from strong volcano activities.

The long-term characteristics of the SAL can be monitored mainly using ground-based lidar observations, balloon-borne in-situ measurements, and space-borne detection (Chouza et al., 2020; Trickl et al., 2024). Lidar is considered a great approach for vertically resolved observation of SALs, providing high spatiotemporal resolution. Long-term ground-based lidar observations are crucial for ensuring the continuity of stratospheric aerosol measurements. There are several long-term (exceeding one decade) datasets of stratospheric aerosols observed by lidar at various locations, including Mauna Loa in Hawaii, US (19.5°N, 156°W, Chouza et al., 2020), NASA Langley Research Center in Hampton, US (37.1°N, 76.3°W, Woods et al., 2003), Garmisch-Partenkirchen in Germany (47.5°N, 11.1°E, Trickl et al., 2013, 2024), São José dos Campos in Brazil (23.2S, 45.9W, Clemesha et al., 1997), Tsukuba in Japan (36.1°N, 140.1°E, Sakai et al., 2016), Lauder in New Zealand (45.0°S 169.7°E, Sakai et al., 2016), Tomsk in Russia (56.48°N, 85.05°E, Zuev, et al., 2017), and the Observatoire de Haute-Provence in Franch (43.9°N, 5.7°E, Khaykin et al., 2017) (see Fig. 1). However, such long-term lidar observations of stratospheric aerosols are still lacking due to insufficient geographical coverage.

Wuhan (30.5°N, 114.4°E) is a central Chinese city located in a transitional region between the tropics and mid-latitude of



the Northern Hemisphere, significantly impacted by the Asian Monsoon in summer. The Asian monsoon anticyclone (AMA)
emerges in response to persistent deep convection over India and Southeast Asia during the boreal summer (Garny and Randel,
2016), controlling the transport patterns of aerosol plumes in the upper troposphere and lower stratosphere (UTLS) over East
Asia. The AMA captures volcanic-emitted stratospheric aerosols to retain and transport at mid-latitudes in Asia (Zhuang and
Yi, 2016; Jing et al., 2023). Moreover, the AMA facilitates efficient vertical transport of tropospheric aerosols to the UTLS
(Garny and Randel, 2016; Yu et al., 2017), forming the so-called 'Asian tropopause aerosol layer' (ATAL), which may also
contribute to stratospheric aerosol levels. Therefore, conducting long-term lidar observations in such a location is highly
valuable.

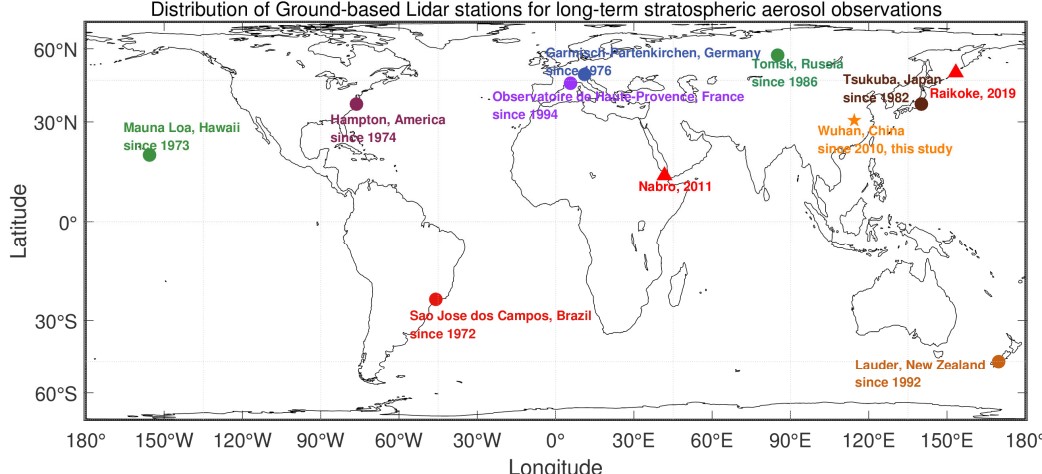

**Figure 1. The locations of ground-based lidar sites with long-term stratospheric aerosol observations and two main volcanic eruptions (Nabro 2011 and Raikoke 2019).**

In this study, we analyze the long-term characteristics of stratospheric aerosols over Wuhan using persistent observations
from a ground-based polarization lidar together with data from several spaceborne instruments during 2010-2021. This paper
is organized as follows. Section 2 provides a brief description of the instruments used and the data processing methods. In Sect.
3, we present the statistical characteristics and significant injection events of stratospheric aerosols. Last, a summary and
conclusions are presented.

## 2.   Instrumentation, data, and methodology

### 2.1 Ground-based polarization lidar at Wuhan

The vertical-resolved optical properties of stratospheric aerosols have been observed with a 532-nm polarization lidar in
Wuhan (30.5°N, 114.4°E) since October 2010 (He et al., 2021, 2022; Yin et al., 2021). A detailed description of the lidar system
can be found in Kong and Yi (2015). Benefiting from the installation of a waterproof transparent roof window, the lidar system
can perform 24/7 routine observations regardless of rainy or snowy conditions since 2017 (Yi et al., 2021). Raw data are stored
with resolutions of 1 minute and 30 meters. The Fernald method (Fernald, 1984) was used to retrieve the backscatter coefficient
$\beta_p$ and aerosol extinction coefficient $\alpha_p$, using a fixed aerosol lidar ratio of 50 sr for the long-term (2010-2021) and 70 sr for
smoke events (Canadian wildfire in 2017 and California wildfire in 2020) (Haarig et al., 2018). The volume depolarization
ratio $\delta_v$ is calculated as the ratio of perpendicular- to parallel-oriented signals, multiplied by the gain ratio, and then converted
into the particle depolarization ratio (PDR) $\delta_p$ (Freudenthaler et al., 2009). The maximum monthly mean tropopause over
Wuhan was approximately 17.0 km during the period from 2010 and 2021, as provided by the Cloud-Aerosol Lidar with



Orthogonal Polarization (CALIOP) Level-3 product. Therefore, the stratospheric aerosol optical depth (sAOD) is calculated by integrating the aerosol extinction coefficient from 17 to 25 km to minimize disturbances from the troposphere and ensure a sufficient signal-to-noise ratio (SNR). The AOD from 25-30 km is approximately $6\times10^{-4}$, as estimated from CALIOP Level-3 data, which would cause a sAOD over Wuhan that is approximately 25% smaller than if integrating from 17 to 30 km. The uncertainties in the derived parameters, as well as the corresponding references, are listed in Table 1.

**Table 1. Estimated uncertainties of the lidar-derived parameters.**

| Parameter | Uncertainty | Reference |
|---|---|---|
| Volume depolarization ratio $\delta_v$ | <5% | Kong and Yi (2015) |
| Particle depolarization ratio $\delta_p$ | 5-10% | Mamouri et al. (2013) |
| Backscatter coefficient $\beta_p$ | <10% | Zhuang and Yi (2016) |
| Extinction coefficient $\alpha_p$ | <20% | Kafle and Coulter (2013) |
| Stratospheric aerosol optical depth, sAOD | 20-25% | Vaughan et al. (2021) |

In addition, an algorithm developed by Yin et al. (2021) was used to screen out all the cloud-free profiles, utilizing a height resolution of 30 meters and a time resolution of 4 hours to ensure a sufficient SNR, with a sliding average of 300 meters for height. The Rayleigh fit method was used to find the reference altitude between 5 and 20 km, with a width of 1.5 km, which the signal is close to the molecular signal derived from meteorological data provided by the Global Data Assimilation System (GDAS1) (Baars et al., 2016). The reference value was set to 0.018 $Mm^{-1}sr^{-1}$ (corresponding to an extinction coefficient of 0.9 $Mm^{-1}$). The molecule backscatter coefficient and extinction coefficient are calculated based on the method presented by Bucholtz (1995), with an uncertainty of <2%. Fernald forward inversion was then applied up to an altitude of 30 km with a lidar ratio of 50 sr (non-smoke) or 70 sr (smoke) to calculate the profiles of the backscatter coefficient and extinction coefficient. Due to the weak signal of stratospheric aerosols, different data processing methods can lead to significant variations in the specific values of the results. From October 2010 to September 2021, 775 nighttime cloud-free profiles were selected for long-term statistical analysis.

Absorption by ozone in the stratosphere was also taken into account. The ozone absorption coefficient is given by:

$$\alpha_{O_3} = \sigma_{O_3} \times n_{O_3} \tag{1}$$

where $\sigma_{O_3}$ =2.72×$10^{-21}$ $cm^2$ is the ozone absorption cross-section (Gorshelev et al., 2014); $n_{O_3}$ is the ozone number concentration (Fig. 1a), provided by the Copernicus Atmosphere Monitoring Service (CAMS) reanalysis data. Figure 2b shows the backscatter coefficient profile with and without correction for ozone absorption on 5 February 2020. The mean residual between 15 and 25 km is approximately $8\times10^{-4}$ $Mm^{-1}sr^{-1}$, which cannot be ignored.

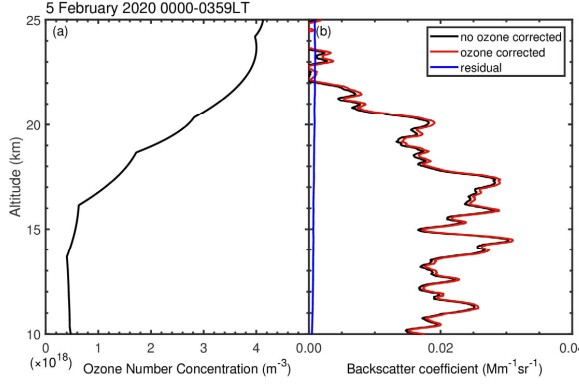

**Figure 2. (a) The ozone number concentration; (b) backscatter coefficient with ozone corrected (red line) and without ozone corrected (black line) over Wuhan at 0000-0359 local time (LT) on 5 February 2020. The blue line represents the residual of the backscatter coefficient after ozone correction.**



### 2.2 CALIOP spaceborne lidar

The space-borne lidar CALIOP, carried on the Cloud-Aerosol Lidar and Infrared Pathfinder Satellite Observation (CALIPSO) satellite, has been widely used to observe the vertically resolved optical and microphysical properties of aerosols and clouds since 2006 (Winker et al., 2007). It is capable of measuring the elastic backscatter at both 532 and 1064 nm, as well as the depolarization ratio at 532 nm near the nadir.

    In this study, CALIOP version 4.10 Level-1B data were used to track smoke plumes from September to October 2017 and
from October to November 2020, illustrating their vertical distributions and optical properties. In addition, CALIOP Level-3 stratospheric aerosol product (Kar et al., 2019) provides monthly mean aerosol optical properties on a spatial grid of 5° in latitude and 20° in longitude. The monthly mean stratospheric aerosol profiles at 32.5°N and 110.0°E were used to estimate the AOD from 25 to 30 km. It should be mentioned that CALIOP Level-3 product consistently exhibits a bit higher aerosol extinction compared with the other satellite-based datasets (Chouza et al., 2020). Moreover, monthly mean tropopause altitudes
from October 2010 to July 2020 were also presented.

### 2.3 OMPS

    The Ozone Mapping and Profiler Suite (OMPS) is installed on the joint NASA/NOAA (National Aeronautics and Space Administration/National Oceanic and Atmospheric Administration) Suomi National Polar-orbiting Partnership (Suomi NPP) satellite, launched in October 2011. OMPS comprises three spectrometers: a downward-looking nadir mapper, a nadir profiler,
and a limb profiler. The UV Aerosol Index (UVAI) is an effective indicator of UV-absorbing aerosols, discerning between aerosol absorption and Rayleigh scattering, which is provided by the nadir-mapper instrument on the Suomi-NPP satellite at a spatial resolution of 50 km × 50 km. The UVAI has been widely employed in detecting elevated aerosols with significant absorption in the atmosphere, such as smoke and dust (Penning de Vries et al., 2009; Lee et al., 2015; Tao et al., 2022). In this study, UVAI provided by the OMPS-NPP Level-2 data product was used to show the horizontal spatial distribution of smoke
plumes.

    Additionally, OMPS-NPP Level-3 data provides aerosol extinction coefficients at a spatial resolution of 5 × 15° lat-lon grid measured by a limb profiler sensor. The monthly mean sAOD at 510 nm at 32.5°N and 112.5°E, provided by OMPS-NPP Level-3 data product, was used to estimate the evolution of Canadian smoke aerosols over Asia from August to November 2017. Furthermore, the monthly mean tropopause altitudes from August 2020 to September 2021 were also presented.

### 2.4 HYSPLIT model

    The NOAA/ARL (National Oceanic and Atmospheric Administration/Air Resources Laboratory) Hybrid Single Particle Lagrangian Integrated Trajectory (HYSPLIT) model (Draxler and Rolph., 2003) can simulate the forward and backward trajectories of an air mass by giving starting time and an initial altitude and geographical location. The meteorological field from the Global Data Assimilation System (GDAS) archive (Kanamitsu, 1989) was used to drive the model in the calculation.
In this study, the simulated backward trajectories were used to track the transport pathway and to confirm the source of smoke plumes.

### 3.    Results and discussions

### 3.1 Overview of stratospheric aerosols

    Figure 3 shows the long-term evolution of height-resolved aerosol backscatter coefficient and sAOD over Wuhan from 2010
to 2021. The monthly mean tropopause altitudes are provided by the CALIOP and OMPS Level-3 products (denoted by the white curve). A stratospheric background aerosol layer, known as the 'Junge Layer', consisting primarily of sulfate from the



oxidation of tropospheric $SO_2$ or $H_2S$ (Junge, 1960; Junge and Manson, 1961), persistently appeared at altitudes of 19.5-23.0 km with an average extinction coefficient of 0.36 $Mm^{-1}$. Three significant episodes with abundant tropospheric aerosols injected are evident: volcanic aerosols from the 2011 Nabro eruption and the 2019 Raikoke eruption, as well as smoke aerosols from the 2017 Canadian wildfire event. Moreover, a weak aerosol plume from the 2020 California wildfire event was also observed.

The stratospheric-quiescent period, characterized by a sAOD of 0.0023 from January 2013 to August 2017, can be defined as reflecting the background level of stratospheric aerosols over Wuhan. Volcanic Explosivity Index (VEI) is a general indicator of the explosive character of a volcanic eruption. It compositely estimates Walker's magnitude, intensity, destructiveness, dispersive power, and energy release rate, and is assigned a value from 1 to 8. Two volcanos with a VEI≥ 4 erupted during this period: Kelud in Indonesia (7.9°S) in February 2014 and Wolf in the Galápagos Islands (0.0°N) in May 2015. Kelud volcanic aerosols were reported to be detected over high latitudes in Tomsk (56.5°N) (Zuev et al., 2017) and the Observatoire de Haute-Provence (OHP) (43.9°N) (Khaykin et al. 2017) in January 2015 due to the meridional aerosol transport. However, no aerosol plumes were observed by our lidar over Wuhan in the first quarter of 2015, and the increase in sAOD during this period cannot be attributed to the influence of Kelud. Several reasons are considered. First, it cannot be completely ruled out that a few Kelud aerosol plumes passed over Wuhan but were not observed due to weather conditions or hardware maintenance. Second, aerosol meridional transport from tropical into extratropical (middle) latitudes generally intensifies during the cold half of the year (October to March of the following year) (Niwano et al., 2009), causing an increase in aerosol optical properties in winter (Sakai et al., 2016; Zuev et al., 2017). A similar increase in sAOD can also be found at the turn of the year (Fig. 1b). A detailed discussion of seasonal characteristics will be presented in Sect. 3.4. Therefore, it cannot be determined whether the increase in sAOD at the beginning of 2015 was due to the influence of Kelud or the seasonal meridional transport of tropical aerosols.

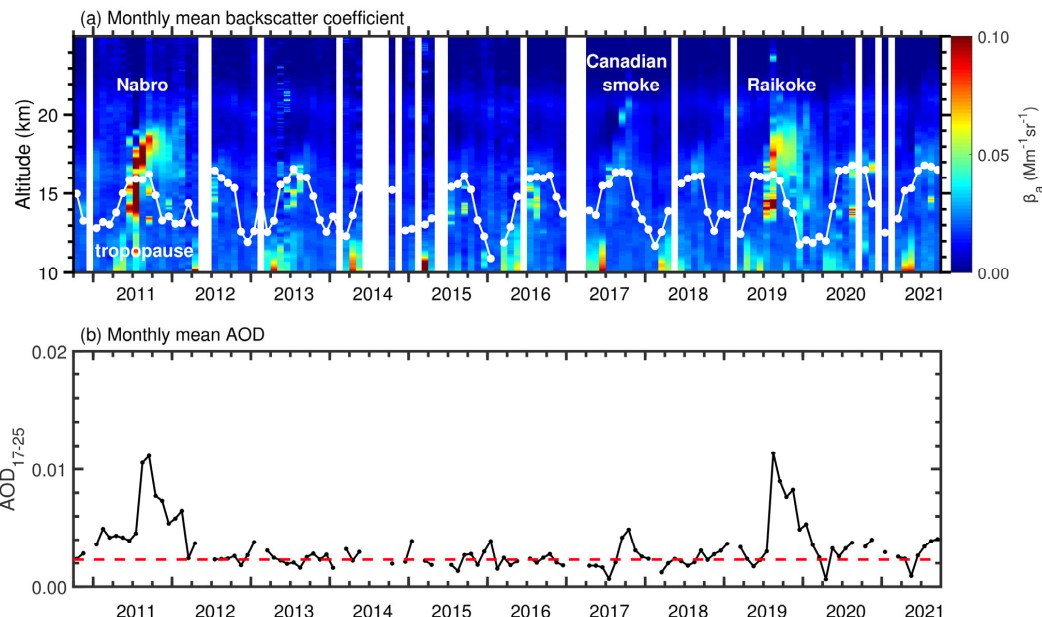

**Figure 3. (a) Time-height contour plots of the aerosol backscatter coefficient measured by 532-nm polarization lidar over Wuhan during 2010-2021; the white curve denotes the monthly mean tropopause from CALIOP (October 2010 to July 2020) and OMPS (August 2020 to September 2021). (b) The evolution of monthly mean 532-nm sAOD from 17.0-25.0 km derived from polarization lidar observation (black curve) at Wuhan. The red dashed line represents the background sAOD of 0.0023.**

**Table 2. Stratospheric aerosol optical depths at different sites worldwide. The sAODs at Tsukuba, Lauder, and Garmisch-**



**Partenkirchen were integrated backscattering coefficient (IBC) by multiplying by a lidar ratio. The period between 2010 and 2021**
**represents the entire lidar measurement period over Wuhan. The period between January 2013 and August 2017 represents the**
**stratospheric-quiescent period.**

| Location | Period | Instrument | Wavelength (nm) | Lidar ratio | AOD ($\times 10^{-3}$) | Integral range (km) | Reference |
|---|---|---|---|---|---|---|---|
| **Wuhan** | **2010-2021** | **Polarization lidar** | **532** | **50** | **3.3** | **17-25** | **this study** |
| **(30.5°N, 114.4°E)** | **2013.1-2017.8** | | | | **2.3** | | |
| Tsukuba (36.1°N, 140.1°E) | 2000-2015 | Lidar systems including the Ruby, Mark-II, Multicolor, Cloud, Continuous First, and Continuous Second lidar | 532 | 50 | 6.0 | Tropopause-33 | Sakai et al. (2016) |
| Lauder (45.0°S, 169.7°E) | 2000-2015 | Lidar system including the First and Second lidar | 532 | 46 | 5.2 | Tropopause-33 | Sakai et al. (2016) |
| Mauna Loa Observatory, Hawaii (19.5°N, 155.6°E) | 1999-2006 | Mauna Loa Stratospheric Ozone Lidar | 532 | 50 | 2.9 | 17-33 | Chouza et al. (2020) |
| | 2006-2013 | | | | 5.0 | | |
| | 2013.1-2019.7 | | | | 4.4 | | |
| Observatoire de Haute-Provence, France (43.9°N, 5.7°E) | 1994-2003 | Differential absorption Lidar; Rayleigh–Mie–Raman lidar | 532 | 50 | 2.4 | 17-30 | Khaykin et al. (2017) |
| | 2013-2014 | | | | 2.8 | | |
| Garmisch-Partenkirchen, Germany (47°N, 11°E) | background in 1979 | Ruby lidar; Water-vapour differential-absorption lidar | 694 | 50 | 2.5 | Tropopause+1~top of the layer (~30km) | Trickl et al., (2013) |

Table 2 lists the lidar-derived sAOD at different sites worldwide. Note that the AOD at Tsukuba, Lauder, and Garmisch-Partenkirchen are integrated backscattering coefficient (IBC) by multiplying a lidar ratio of 50 or 46 sr. The background sAOD over Wuhan of 0.0023 was slightly smaller because it only contained aerosols below 25 km instead of 30 km. As mentioned
in Sect. 2, AOD between 25 and 30 km was estimated to be approximately 0.0006 based on CALIOP data. Due to the weak signal of stratospheric aerosols, different data processing methods can lead to variations in the exact sAOD values. Furthermore, it is noteworthy that the sAOD during the stratospheric-quiescent period over OHP increased from 0.0024 in 1997-2003 to 0.0028 in 2013-2014, revealing the significant contribution of volcanic aerosols in the first decade of the 21st century. Several volcanos with VEI≥ 4 erupted in the tropics and Northern Hemisphere during this period, such as Shiveluch in 2001 (Thomason
and Pitts, 2008), Okmok in 2008 (Bazhenov et al., 2012), and Eyjafjallajökull in 2010 (Sicard et al., 2012).

The volcanic aerosol layers from the Nabro and Raikoke eruptions can be observed at altitudes 15-25 km during the second half of 2011 and 2019, respectively (Zhuang and Yi, 2016; Jing et al., 2023). An enhanced sAOD of 0.011 was observed during both the Nabro and Raikoke events, which was 4.8 times the background sAOD (0.0023). Another enhancement of the backscatter coefficient appeared around 20 km after September 2017, caused by the smoke aerosol injection from the Canadian
wildfire event in August 2017. The intense pyro-cumulonimbus (PyroCb) released approximately 0.1-0.3 Tg aerosols into the low stratosphere, comparable in quantity to those emitted from a moderate volcanic eruption (Peterson et al., 2018). An increase in sAOD in 2017 (0.0049) was 2.1 times the background sAOD (0.0023). The smoke plumes over Wuhan will be analyzed further in Sect. 3.3.

In addition, stratospheric aerosols are contributed by other tropospheric sources. Periodic increases in the backscatter
coefficient were observed below an altitude of 18 km during the summer. The majority of these aerosol layers existed below the tropopause and are known as the ATAL (Vernier et al., 2015; Yu et al., 2015). The AMA emerges in response to persistent deep convection over India and Southeast Asia during the boreal summer (Garny and Randel, 2016), leading to efficient vertical transport from the surface to the UTLS (Baker et al., 2011). Due to extensive human activities and the influence of the AMA, the ATAL forms and exists in the UTLS at altitudes of 13-18 km over megacities, consisting mainly of sulfate and carbonaceous
aerosols (Vernier et al., 2015). The observation of the ATAL confirmed that anthropogenic aerosols and/or their gas-phase precursors can be transported to the UTLS, although they are generally removed effectively through precipitation, according



to previous understanding (Mari et al., 2010).

### 3.2 Volcanic aerosol plumes

### 3.2.1 Nabro volcanic aerosols in 2011

Nabro volcano (13.4°N, 41.7°E) erupted on 12 June 2011, and the emitted volcanic aerosols were transported eastward to Wuhan from 19 June onward (Zhuang and Yi, 2016), as shown in Fig. 4. The volcanic aerosols persistently appeared over Wuhan at altitudes of 16-20 km until October 2011. During the initial stage, the Nabro aerosol plume exhibited a mean $\beta_p$ of 0.33 Mm$^{-1}$sr$^{-1}$ at 16.0-19.3 km on 8 July, and 0.07 Mm$^{-1}$sr$^{-1}$ at 17.0-19.0 km on 12 July. Such strong variability in both the backscatter coefficient and vertical distribution suggests an inhomogeneous spatial density of the aerosol plume. After August

2011, the Nabro aerosol plume generally diffused over a wider altitude range of 15.0-20.0 km, with a smaller mean $\beta_p$ of <0.06 Mm$^{-1}$sr$^{-1}$. Moreover, the integrated backscatter coefficient steadily decreased from mid-August to December, yielding an e-folding decay time of ~130 days, as reported by Zhuang and Yi (2016).

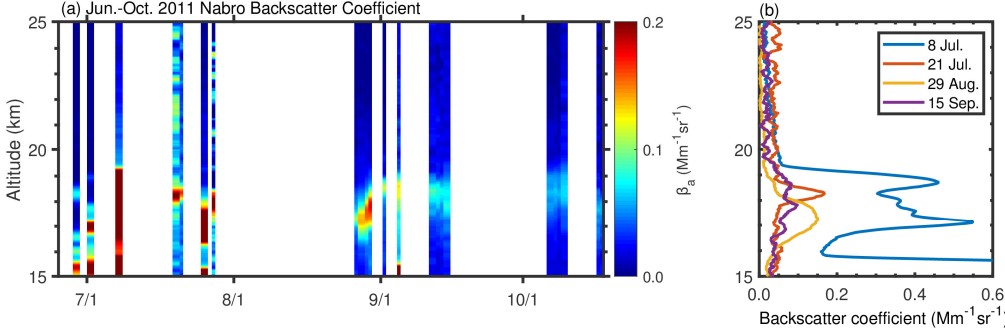

**Figure 4. (a) The nighttime backscatter coefficient of volcanic aerosols of Nabro from June to October 2011. White strips denote that**
**data are unavailable due to unfavorable weather conditions or hardware maintenance. (b) The backscatter coefficient $\beta_p$ profiles of Nabro aerosol plume derived from polarization lidar observations over Wuhan.**

### 3.2.2 Raikoke volcanic aerosols in 2019

Raikoke volcano (48.3°N, 153.3°E) erupted on 21-22 June 2019, resulting in two types of volcanic aerosol plumes, i.e. the main aerosol plume and a small but impacted aerosol cloud known as 'Coherent Circular Cloud' (CCC), as shown in Fig. 5.

Jing et al. (2023) studied the transport pathway of the Raikoke volcanic aerosols and their optical properties over Wuhan. The main aerosol plume was initially transported eastward across North America, the Atlantic, and Europe before mid-July, eventually reaching south of Sakhalin Island on 22 July. Driven by the AMA, the transport pathway then turned southwestward, arriving in Wuhan on 25 July, having a mean $\beta_p$ of 0.04 Mm$^{-1}$sr$^{-1}$. This main aerosol plume intermittently diffused at 15.0-20.0 km over Wuhan in the following months. Observations showed a decrease in layer-mean $\beta_p$ from 0.16 Mm$^{-1}$sr$^{-1}$ on 2 August

to 0.04 Mm$^{-1}$sr$^{-1}$ on 23 September over Wuhan.

In addition, another impacted volcanic aerosol plume with a narrow horizontal extent of approximately 300 km, known as CCC, first appeared at 19.0-20.0 km near Kamchatka, Russia on 18 July. It then moved southward to latitudes between 20°N and 30°N, completing three full-circle transports around the Earth over the following two months (Chouza et al., 2020; Gorkavyi et al., 2021). Interestingly, as shown in Fig. 5a, the first two circles were observed by our polarization lidar when the

CCC passed over Wuhan, at approximately 21.0 km on 30 July and at around 24.0 km on 24 August, indicating the presence of self-lofting during the 25 days. The peak $\beta_p$ of CCC was measured to be 6.5 Mm$^{-1}$sr$^{-1}$ on 30 July and 2.0 Mm$^{-1}$sr$^{-1}$ on 24



August, an order of magnitude larger than of the main aerosol plume.

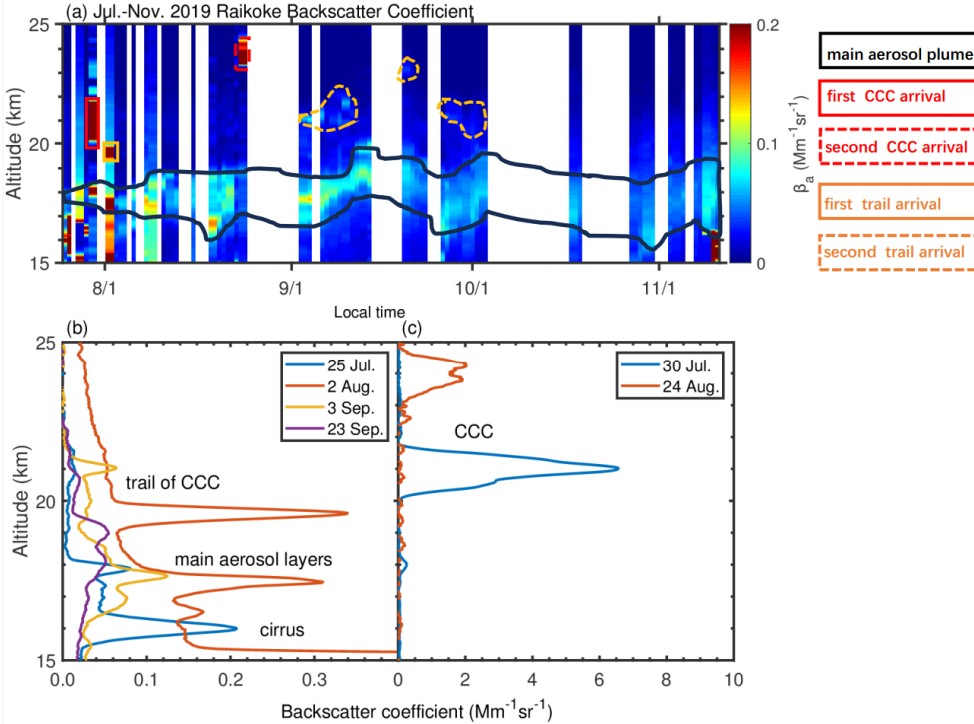

**Figure 5. (a) The nighttime backscatter coefficient of volcanic aerosols of Raikoke from July to November 2019. White strips denote**
**that data are unavailable due to weather conditions or hardware maintenance. The backscatter coefficient $\beta_p$ profiles of (b) the main**
**aerosol plume and (c) CCC derived from polarization lidar observations over Wuhan.**

### 3.3 Smoke aerosol plumes

#### 3.3.1 Canadian wildfire smoke in 2017

In August 2017, severe wildfires occurred in western Canada and the northwestern United States (Peterson et al., 2018).
Large amounts of smoke were rapidly lifted to the UTLS (within less than an hour) through the formation of pyro-
cumulonimbus clouds. A portion of the smoke plume that separated from the initial emission was transported to midlatitudes
and later was observed over Wuhan in September 2017. The detail of the transport of this smoke plume to Wuhan is described
in Appendix A. Figure 6 shows the backscatter coefficient and depolarization ratio derived from polarization lidar observations
over Wuhan from September to October 2017. The white curve denotes the altitude of the tropopause. Smoke aerosol plumes
were first observed on 14-17 September at altitudes of 19.0-21.0 km, with a mean $\beta_p$ of 0.05-0.10 Mm$^{-1}$sr$^{-1}$. The $\delta_p$ values of
0.14-0.18 were relatively large, suggesting that the smoke layer was composed of irregularly shaped, dry, and non-coated soot
particles (Ansmann et al., 2018; Ohneiser et al., 2020). One and a half months later, two distinct smoke layers appeared at
altitudes of 20.3-21.5 km and 22.1-23.0 km, with a mean $\beta_p$ of 0.04 Mm$^{-1}$sr$^{-1}$ and $\delta_p$ of 0.16. Compared with the mid-
September measurements, the smaller $\beta_p$ indicates the dissipation of smoke plumes over time. However, $\delta_p$ remained almost
unchanged, confirming that the aging process of smoke particles is rather slow in the stratosphere as compared with that in the
troposphere. Chemical interactions with trace gases in the troposphere are more likely to alter the shape of smoke particles
(China et al., 2015). The plume-isolated AOD from September to October was 0.002-0.010, at least an order of magnitude
larger than the background sAOD (0.0023) over Wuhan.





As a comparison, Canadian wildfire smoke measured over OHP, France from 24 August to 26 September showed a similar

(to Wuhan) $\delta_p$ of approximately 0.15 (Khaykin et al., 2018). However, the plume-insolated AOD for this smoke event is much

larger in Europe. A plume-isolated AOD of up to 0.7 was measured over OHP, France on 29 August, which was two orders of

magnitude larger than that of 0.002-0.010 over Wuhan (Khaykin et al., 2018). Similarly, the layer-integrated AOD of the smoke

plume reached 0.3 in the free troposphere and 0.6 in the stratosphere over Kosetice, Czech Republic on 22 August (Ansmann

et al., 2018). Measurements of smoke aerosols over Europe based on the European Aerosol Research Lidar Network

(EARLINET) showed larger sAOD values from >0.25 on 21-23 August to 0.005-0.030 on 5-10 September (Baars et al., 2019).

This is because the Canadian smoke mainly remained and was transported at high latitudes in the first two months after the

wildfire event.

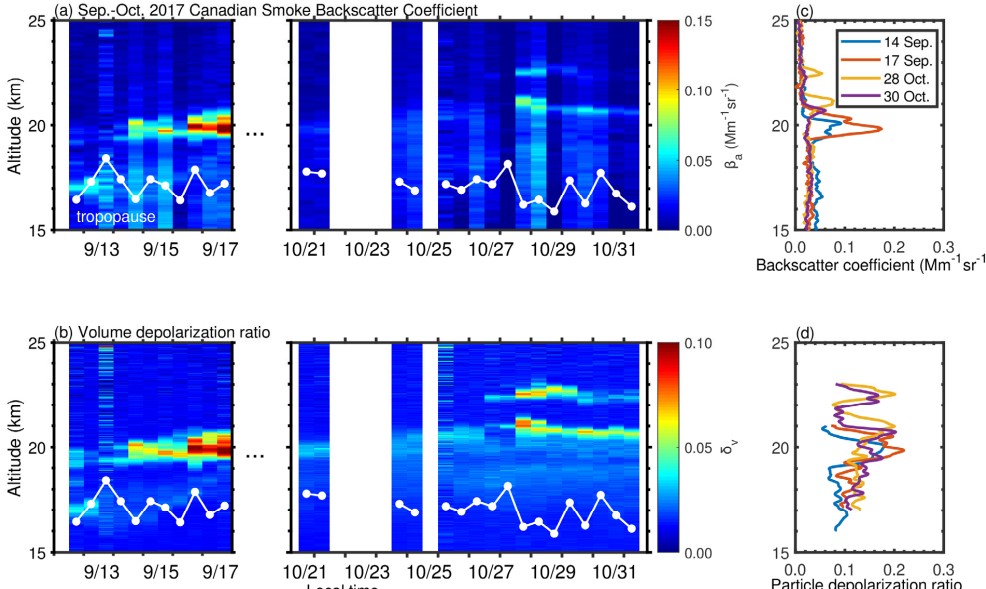

**Figure 6. Four-hour integrated nighttime (a) particle backscatter coefficient $\beta_p$, and (b) volume depolarization ratio $\delta_v$ measured**

**by polarization lidar over Wuhan from September to October 2017. White strips denote that data are unavailable due to weather**

**conditions or hardware maintenance. The white curve denotes the altitudes of the tropopause. The (c) aerosol backscatter coefficient**

**and (d) particle depolarization ratio profiles derived from polarization lidar observations.**

Using the Chemical Lagrangian Model of the Stratosphere (CLaMS) model, Kloss et al. (2019) found that the fire plume

initialized on 12-14 August over western Canada, and was transported eastward at latitudes north of 40°N. The plumes passed

over Europe in early- to mid-August and reached the Asian monsoon area at the end of August, with a fraction moving to low

latitudes along the eastern flank of the AMA. When the AMA broke down in September, the smoke plume had spread

throughout the Northern Hemisphere. Previous studies have shown that stratospheric aerosols at high latitudes in the Northern

Hemisphere can be transported to the middle and low latitudes via the AMA (Kloss et al., 2021; Jing et al., 2023). In Fig. 7,

the OMPS monthly mean sAOD generally increased from a background level of 0.004 to >0.010 to the north of 40°N and

0.007 near Wuhan (30.5°N) in September. Subsequently, the sAOD near Wuhan decreased to 0.006 in October and 0.005 in

November. However, the spatial distribution of sAOD confirms that the smoke aerosols mainly remained at high latitudes with

sAOD values exceeding 0.010.





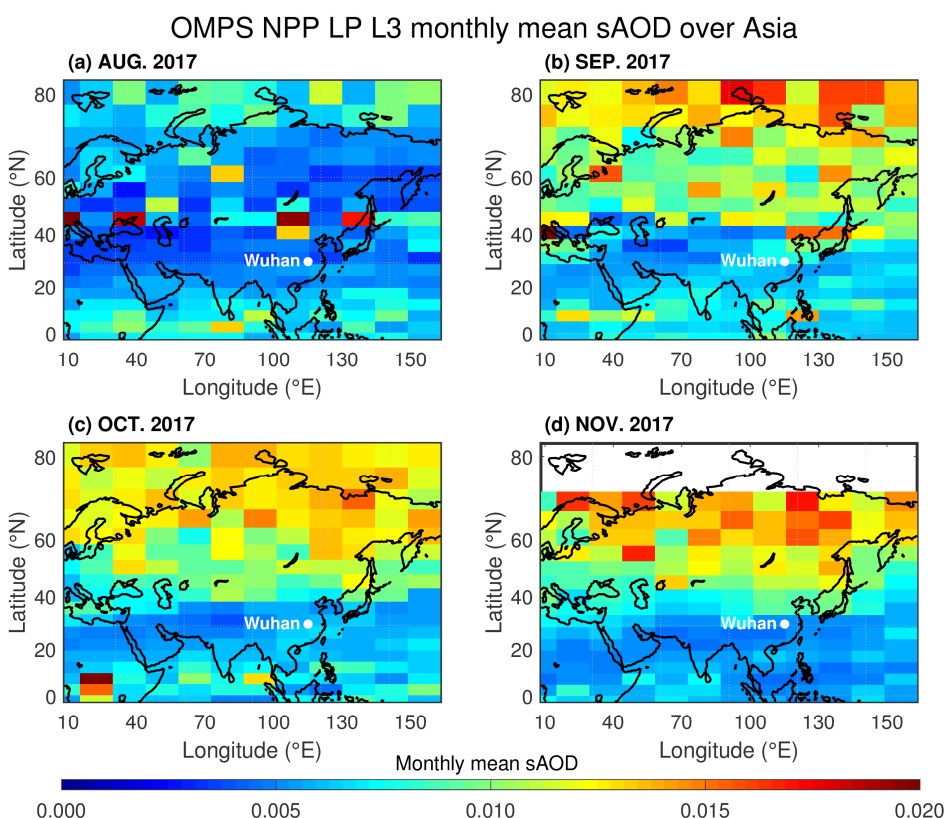

**Figure 7. OMPS-NPP Level-3 monthly mean sAOD from August to November 2017.**

### 3.3.2 Californian wildfire smoke in 2020

In October 2020, record-breaking wildfires occurred in California, emitting a significant amount of smoke into the UTLS, which was then transported eastward (Safford et al., 2022). Figure 8 shows the polarization lidar observation over Wuhan on 8-9 November 2020. A thin aerosol layer was observed above the tropopause at 16.0-17.0 km, with a mean $\beta_p$ of 0.11 Mm$^{-1}$sr$^{-1}$ and a mean $\delta_p$ of 0.13, indicating that the aerosol layer mainly consisted of non-spherical smoke particles. Rapid lofting into the dry upper troposphere prevents the aging of the smoke particles (Baars et al., 2019). In Cyprus, smoke aerosols from this Californian wildfire were observed earlier on 27 October at 11.0-13.0 km, with $\beta_p$ values of 1-3 Mm$^{-1}$sr$^{-1}$ (Mamouri et al., 2023), an order of magnitude larger than that observed in Wuhan. The $\delta_p$ measured in Cyprus was 0.10-0.15, consistent with that over Wuhan (0.13), indicating the slow aging process of smoke aerosols in the stratosphere. The detail of smoke plume transport to Wuhan is described in Appendix B.



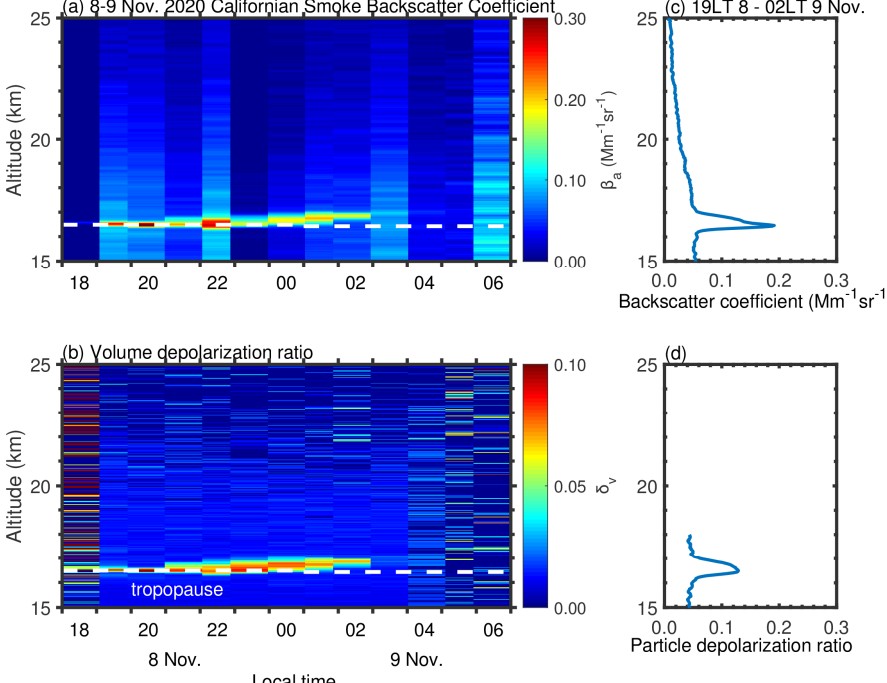

**Figure 8.** One-hour integrated nighttime (a) particle backscatter coefficient $\beta_p$, and (b) volume depolarization ratio $\delta_v$ derived from polarization lidar over Wuhan on 8-9 November 2020. White strips denote that data are unavailable due to weather conditions or hardware maintenance. The white curve denotes the altitudes of the tropopause. The profiles of the (c) aerosol backscatter coefficient and (d) particle depolarization ratio derived from polarization lidar observations are also provided.

### 3.4 Seasonal variation

Understanding the seasonal variations in stratospheric aerosol patterns is crucial for gaining deeper insights into their feedback on weather and climate. Figure 9a shows the profiles of the seasonal mean backscatter coefficient $\beta_p$ over Wuhan for each season during the stratospheric-quiescent period from January 2013 to August 2017 as well as during the whole measurement period. Here the four seasons are defined as: spring (March-April-May), summer (June-July-August), autumn (September-October-November), and winter (December-January-February). The seasonal mean tropopause heights were 13.7 km in spring, 16.1 km in summer, 15.0 km in autumn, and 13.2 km in winter. At altitudes of 19.5-23.0 km, an enhancement of aerosol extinction was observed in all seasons with a mean $\beta_p$ of $7.6\times10^{-3}$ Mm$^{-1}$sr$^{-1}$. This non-seasonal layer, known as the 'Junge Layer', is a global aerosol layer at around 20 km altitude (Junge, 1960; Junge and Manson, 1961).

In summer, another distinct aerosol layer was observed at 13-28 km, with a mean $\beta_p$ of 0.023 Mm$^{-1}$ sr$^{-1}$, approximately 1.4-1.6 times larger than those in other seasons. This layer was contributed by ATAL. The AOD of ATAL at 13.0-18.0 km over Wuhan was 0.0057, consistent with an increasing AOD of 0.002-0.006 at 13.0-18.0 km in the entire ATAL region (5-105°E and 15-45°N) during 1995-2013 (Veriner et al., 2015). The AMA facilitates efficient vertical transport from the surface to the UTLS, serving as a primary source of young air in the lower stratosphere (Randel et al., 2010) and bringing anthropogenic aerosols and/or their gas-phase precursors from the lower troposphere to the UTLS. Furthermore, injections of aerosols from volcano eruptions and wildfires caused the larger $\beta_p$ of 0.019 Mm$^{-1}$sr$^{-1}$ at 16.0-20.0 km in summer and autumn (compared with 0.016 Mm$^{-1}$sr$^{-1}$ in winter and 0.012 Mm$^{-1}$sr$^{-1}$ in spring), because Wuhan was generally affected by those events during the summer and autumn months by summer monsoon circulation, which promotes effective mixing between the extratropics and





tropics and influence the tropical seasonal cycle of different atmospheric components (Abalos et al., 2013).

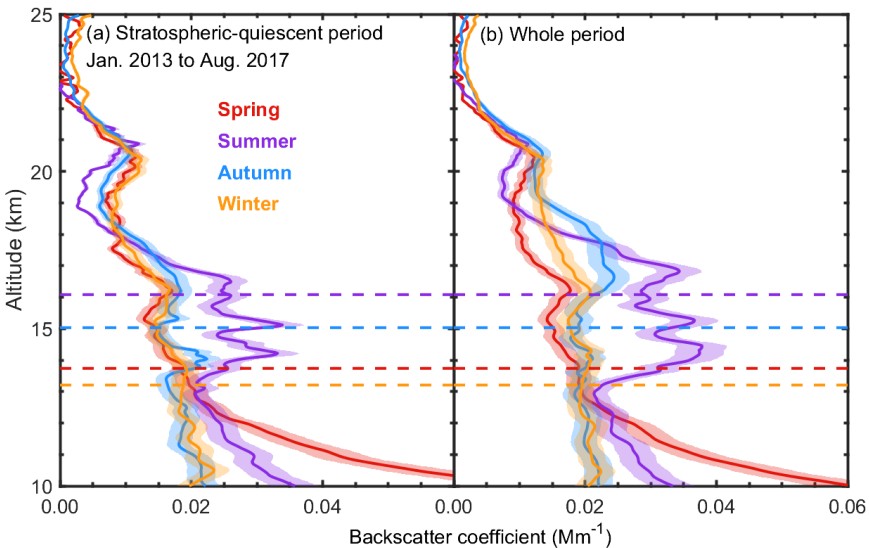

**Figure 9. Profiles of seasonal mean aerosol backscatter coefficient $\beta_p$ during (a) the stratospheric-quiescent period from January 2013 to June 2019 and (b) the entire period from 2010 to 2021. The dashed lines represent the tropopause for each season.**

In Fig. 10, we present the differences in the mean sAOD between the cold half-year (October-next March) and warm half-year (April-September). The mean sAOD in the cold half-year was 0.0026, approximately 24% larger than that of 0.0021 in the warm half-year. Because aerosols transported meridionally from the tropics to middle and high latitudes generally

intensified the cold half-year in the Northern Hemisphere (Niwano et al., 2009). This process provides additional aerosol mass from the stratospheric tropical aerosol reservoir. Similarly, Zuev et al., (2017) found that the integrated backscattering coefficient was larger over Tomsk, Russia, during the cold half-year from 2000 to 2016. This pattern was also observed in Tsukuba, Japan, where the stratospheric aerosol backscatter ratio (R-1) was 40% larger in winter than in summer at 22.0-23.0 km from 1997 to 2004 (Sakai et al., 2016).

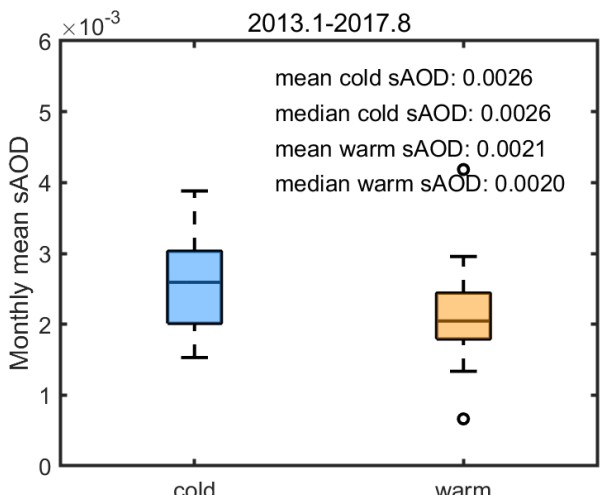

**Figure 10. Mean sAOD in the cold half-year (October-next March) and warm half-year (April-September).**



### 3.5 Radiative forcing by stratospheric aerosols

Stratospheric aerosols modify radiative fluxes by scattering and absorbing solar radiation. Hence amounts of aerosols emitted from those great volcanic eruptions, such as the 1815 Tambora eruption and 1991 Pinatubo eruption, can significantly cool the global climate (Solomon et al., 2011). Figure 11 shows the estimated radiative forcing (RF) induced by stratospheric aerosols and sAOD over Wuhan from 2010 to 2021. Radiative forcing was calculated by multiplying the sAOD by a conversion factor of -25 W·m$^{-2}$ (black points) (Hanson et al., 2005; Solomon et al., 2011).

The radiative forcing of smoke is very complicated, depending on the surface albedo as well as the composition (Heinold et al., 2022). Black carbon (BC) in wildfire smoke generally exhibits strong absorption of solar radiation, unlike sulfate aerosols which reflect solar energy back to space, leading to different climate responses. Therefore, the conversion factor from sAOD to RF should be re-estimated to obtain a more reliable RF. The organic carbon (OC) and BC emissions from biomass burning are generally proportional. According to Koch et al. (2001), the organic matter (OM) to BC mass ratio (OM/BC) is set to be 7.9 and the OC to OM mass ratio (OC/OM) is assumed to be 1.3. As estimated by Hanson et al. (2005), the conversion factors from AOD to RF are -13 W·m$^{-2}$ for OC and 60 W·m$^{-2}$ and BC. As a result, the contribution of sAOD to RF can be divided into three parts: background sAOD (sAOD$_{background}$), OC sAOD (sAOD$_{OC}$) and BC sAOD ($s$AOD$_{BC}$). The RF during smoke injection period can be calculated as follows:

$$RF_{smoke} = sAOD_{background} \times (-25) + sAOD_{OC} \times (-13) + sAOD_{BC} \times 60 \qquad (2)$$

where the sAOD$_{background}$ is 0.0023, as given in Sect. 3.1. The corrected RF during smoke intrusion period is presented (red points) in Fig. 11. After the correction, smoke RF becomes slightly weaker.

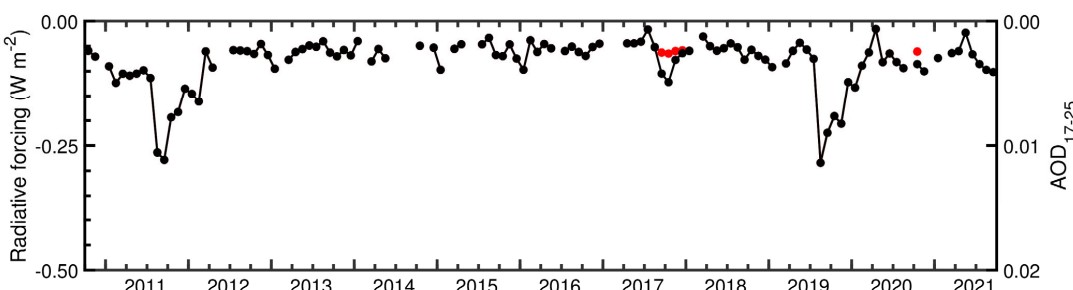

**Figure 11. Temporal variations of stratospheric aerosol radiative forcing and sAOD over Wuhan. The red points represent the corrected radiative forcing caused by smoke aerosols.**

The RF values mainly range from -0.04 W·m$^{-2}$ to -0.28 W·m$^{-2}$ with a mean of -0.08 W·m$^{-2}$. The 2011 Nabro and 2019 Raikoke eruptions caused remarkable cooling effects as seen from the RF of -0.28 W·m$^{-2}$. The stratospheric-quiescent period shows a stable trend of RF around -0.05 W·m$^{-2}$, representing the stratospheric background aerosol level over Wuhan. On the contrary, Solomon et al., (2011) found that global stratospheric aerosols have been increasing by ~7% year$^{-1}$ based on satellite observations in the previous decade from 2000 to 2010. The abundant volcano activities caused a significant increase in stratospheric aerosols. There were 12 eruptions with VEI≥ 4 in the tropics and Northern Hemisphere before 2010 statistics by Chouza et al. (2020). While only three volcanos erupted with VEI≥ 4 after 2010, i.e., the Nabro in Eritrea in 2011, the Kelud in Indonesia in 2014, and Wolf in the Galápagos Islands in 2015.

The mean RF during smoke observation periods was -0.06 W·m$^{-2}$, consistent with the conclusion drawn by Hanson et al. (2005) that biomass burning (BC + OC) shows a negative forcing. This value was close to the background level of stratospheric aerosols and resulted in weaker negative forcing compared with sulfate aerosols. This is due to the absorption of solar radiation by BC, which offsets a portion of solar radiation reflected by OC and background aerosols.

For comparison, Chouza et al. (2020) measured a close value of -0.11 W·m$^{-2}$ over Mauna Loa Observatory, Hawaii during



a volcanic quiescent period in 2013-2019. Khaykin et al. (2017) measured the background value of -0.06 W·m$^{-2}$ over Observatoire de Haute-Provence, France during 1997-2003. As mentioned above, the RF derived from sAOD over Wuhan only contained aerosols below 25.0 km instead of 30.0 km. Therefore, approximately 0.0006 for sAOD or -0.015 W·m$^{-2}$ for RF was estimated between 25.0 and 30.0 km based on CALIOP data. Meanwhile, different data processing methods can lead to significant variations in the specific values of the results due to the weakness of stratospheric aerosols.

### 4. Summary and conclusions

This study analyzes the long-term characteristics of stratospheric aerosols over Wuhan from 2010 to 2021, mainly using ground-based polarization lidar observations in conjunction with several satellite observations. The eruptions of the volcano Nabro in 2011 and Raikoke in 2019 increased the sAOD by a factor of 4.8 compared to the stratospheric-quiescent period (January 2013 to August 2017). The sAOD during this quiescent period was 0.0023, reflecting the background level of stratospheric aerosols over Wuhan, consistent with previous studies during a similar period (Khaykin et al., 2017; Chouza et al., 2020).

We also presented observations of the volcanic aerosol layers from the Nabro eruption in 2011 and the Raikoke eruption in 2019 over Wuhan, which have been discussed in detail in our previous studies (Zhuang and Yi, 2016; Jing et al., 2023). In late August 2017, a historically severe wildfire in western Canada emitted large amounts of smoke to the UTLS; a portion of the smoke plume was transported to Wuhan in September 2017. Two layers with enhanced aerosol extinction were observed: the first with a mean $\beta_p$ of 0.05 Mm$^{-1}$sr$^{-1}$ at 19.0-20.5 km on 14 September, and the second with a mean $\beta_p$ of 0.04 Mm$^{-1}$sr$^{-1}$ at 20.3-23.0 km on 28 October. The $\delta_p$ values were 0.14-0.18, suggesting the composition of irregularly shaped, dry, and non-coated soot particles. The plume-isolated AODs were 0.002-0.010. Additionally, smoke plumes from the Californian wildfire in October 2020 appeared over Wuhan at 16.1-17.1 km on 8-9 November 2020, with a mean $\beta_p$ of 0.11 Mm$^{-1}$sr$^{-1}$ and $\delta_p$ of 0.13.

Seasonal variations were also studied. The ATAL at 13.0-18.0 km showed a mean $\beta_p$ of 0.023 Mm$^{-1}$sr$^{-1}$, 1.4-1.6 times larger than in other seasons during the stratospheric-quiescent period. The mean AOD of the ATAL was 0.0057, confirming that anthropogenic aerosols are an important source of UTLS aerosols. Volcanic aerosols primarily enhanced the mean stratospheric $\beta_p$ in both summer and autumn (0.019 Mm$^{-1}$sr$^{-1}$), which were relatively smaller in winter (0.016 Mm$^{-1}$sr$^{-1}$) and spring (0.012 Mm$^{-1}$sr$^{-1}$) as the volcanic aerosols dissipated. The mean sAOD during the cold half-year (0.0026) was 24% higher than during the warm half-year (0.0021), indicating stronger meridional transport of stratospheric aerosols from the tropics to middle and high latitudes.

The long-term stratospheric aerosol radiative forcing over Wuhan is also presented, revealing the cooling effect caused by stratospheric aerosols. The mean radiative forcing was -0.08 W·m$^{-2}$ during the entire period and -0.05 W·m$^{-2}$ during the stratospheric-quiescent period. The 2011 Nabro and 2019 Raikoke eruptions resulted in a significant cooling effect of -0.28 W·m$^{-2}$.

Figure 12 shows the conceptual diagram of the transport of stratospheric aerosols over East Asia. Wuhan is located in a region significantly impacted by the Asian monsoon during June to September. The AMA captures long-range transported stratospheric aerosols from volcanic eruptions at mid-latitudes. Stratospheric aerosols are then transported along the eastern flank of the AMA, increasing the sAOD over Wuhan (Zhuang and Yi, 2016; Jing et al., 2023). Additionally, the ATAL is facilitated by the vertical transport of tropospheric aerosols to 13.0-18.0 km and sustained by the convective activity of the Asian monsoon (Garny and Randel, 2016; Yu et al., 2017). Canadian smoke was transported to Wuhan via a weakening AMA in September 2017. Due to the breakup of the AMA after September, smoke plume can move directly to Wuhan (e.g., Californian wildfire smoke in November 2020).



Owing to the persistent operation of polarization lidar from 2010 to 2021, we have developed a comprehensive sketch of the long-term characteristics of stratospheric aerosols over Wuhan, including the variations in sAOD, significant stratospheric injection events, seasonal variations, and radiative forcing. This study is an important supplement to stratospheric aerosol measurements in mid-latitude areas, particularly over East Asia, where human activities are abundant. Additionally, the

analyses help us better understand how stratospheric aerosols respond to regional and global climate change (Solomon et al., 2011). However, there are some limitations to our study. Further long-term observation is necessary to achieve continuous monitoring during the stratospheric-quiescent period. The impact on ozone depletion during stratospheric aerosol injection events is also an essential topic (Ohneiser et al., 2022). In addition, more accurate aerosol extinction and lidar ratio can be obtained through our pure rotational Raman lidar (Liu et al., 2019). The system with larger telescope will also improve the

signal-to-noise ratio (SNR) at higher altitudes, complementing extinction measurements above 25 km.

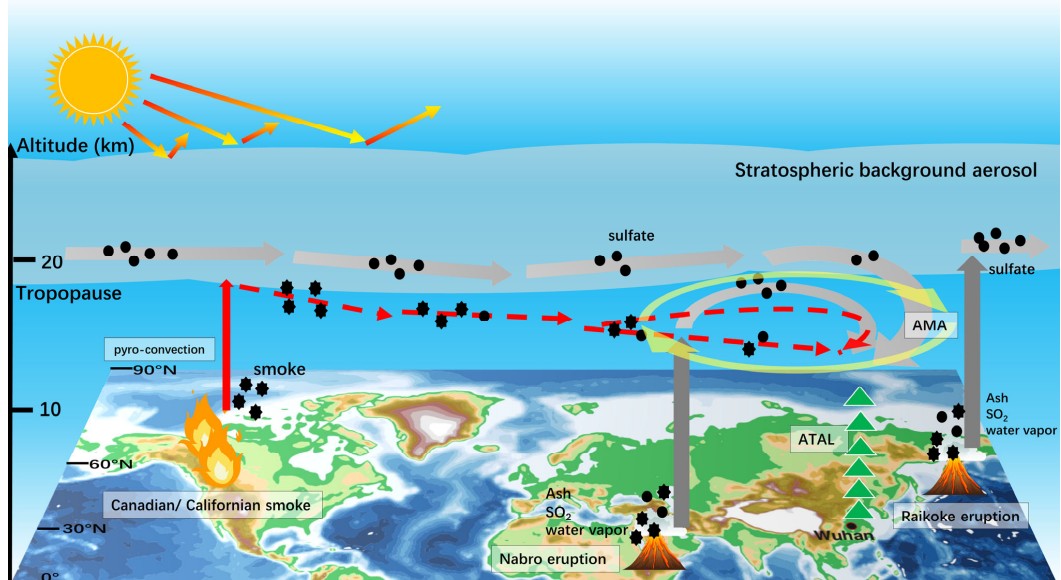

**Figure 12. The conceptual diagram of the transport of stratospheric aerosols over East Asia.**

**Appendix A**

Figure A1 shows the main trajectories of smoke plumes transported in the Northern Hemisphere between 16 August and 21

October 2017 based on CALIOP observations. A large PyroCb plume, referred to as plume O, developed and reached the lower stratosphere on 12-13 August over Canada (Peterson et al., 2018; Hu et al., 2019; Sicard et al., 2019; Torres et al., 2020; Das et al., 2021; Lestrelin et al., 2021). Plume O was transported eastward crossing the Atlantic, reaching Western Europe on 27 August (yellow line). It then split into three plumes due to the wind shear prevailing in the associated jet stream (Lestrelin et al., 2021). We identified their transport pathways using CALIOP observations, denoting them as plume I (dark red), plume II

(blue), and plume III (green).

From September to mid-October, plume II (blue) and III (green) moved eastward between 40-60°N, completing a full circle. Plume I (dark red) moved eastward to Central Asia, turned south to the mid-latitudes in early September, and then moved westward generally along 30°N, completing three-quarters of a circle to the east coast of China by 21 October. In addition, plume IV (purple) was first observed over the Central North Pacific by CALIOP on 3 October, moving westward and

approaching plume I southeast of Japan on 19 October. However, tracking plume IV before 3 October was difficult. We speculate that plume IV separated from plume II in early October due to wind shear.



The sources of two periods of smoke plumes observed over Wuhan were tracked using CALIOP observations and the HYSPLIT model, as shown in Fig. A2. The September plume originated from plume II, observed at altitudes of 18.0-21.0 km with a central coordinate at 59.5°N, 35.9°E on 4 September by CALIOP (Fig. A2(a)). Part of the plume II was transported southeastward to Wuhan on 15 September, based on a 13-day backward trajectory simulation (Fig. A2(c-d)). Meanwhile, an elongated aerosol layer at 20 km observed by CALIOP near Wuhan on 17 September (Fig. A2(b)) confirmed that smoke aerosols had been transported to mid-latitudes. The October Plume came from plume I and plume IV, observed at 20.8 km and 23.1 km, respectively, on 19 October south of Japan (Fig. A2(e)). They continued moving westward together and were observed over Wuhan at altitudes of 22.8 km and 21.1 km, respectively.

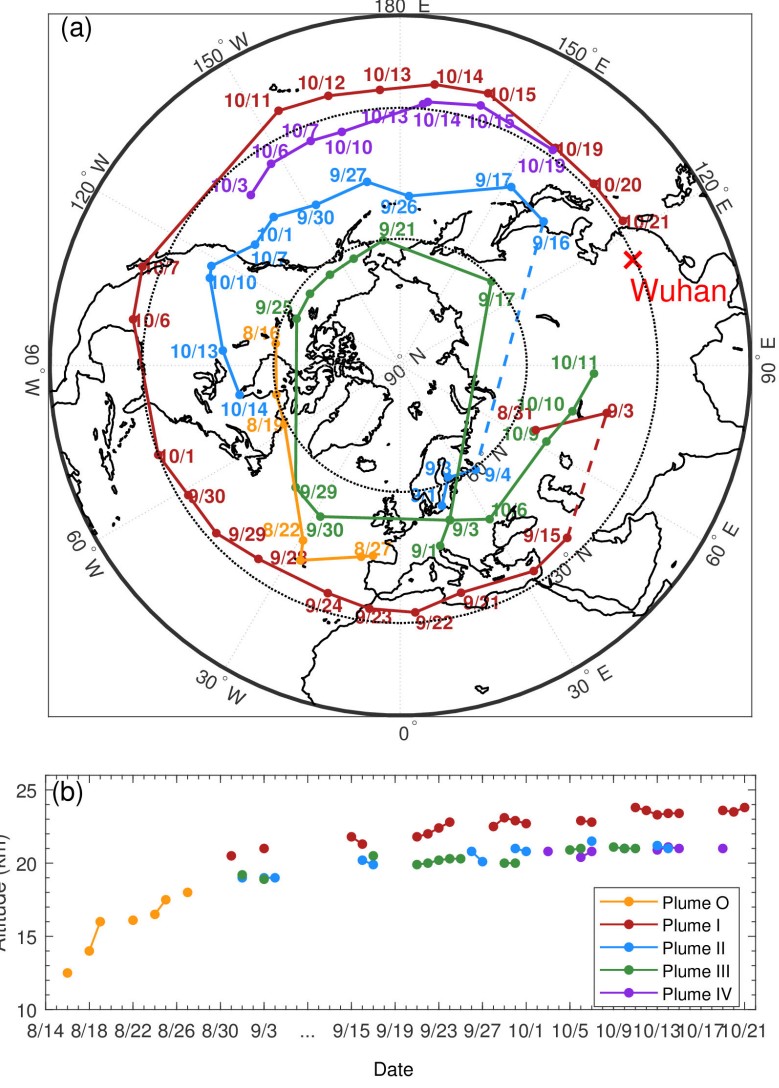

Figure A1. (a) Transport pathways and (b) altitude evolution of smoke plumes between 16 August and 21 October 2017 based on CALIOP observations. CALIOP observations were unavailable on 5-14 September due to solar activity; thus, the potential vorticity tracking in Lestrelin et al., (2021) was shown instead with the dashed lines.



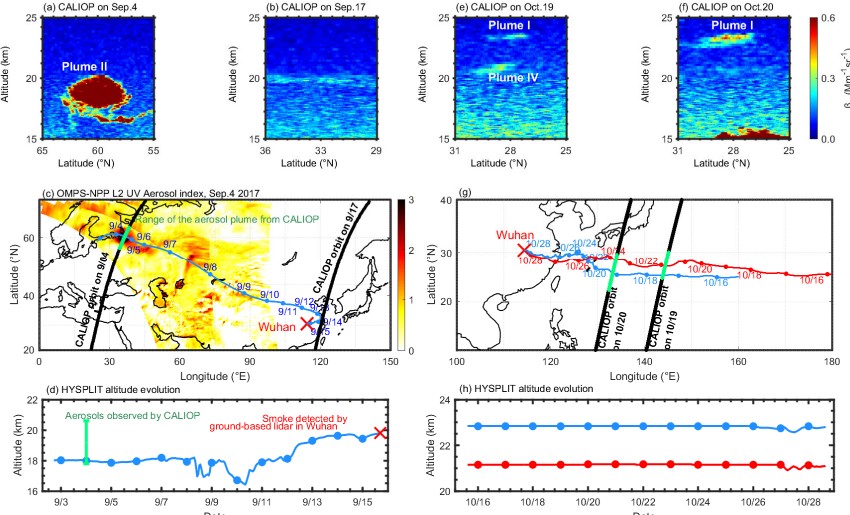

Figure A2. CALIOP-observed 532-nm total attenuated backscatter coefficients on (a) 4 September, (b) 17 September, (e) 19 October, and (f) 20 October 2017. (c) UVAI measured by OMPS on 4 September. The backward trajectories starting from Wuhan on 15 September and 16 October are shown in (c), (d), (g), and (h). The CALIOP footprints in (a) and (b) are shown in (c); the CALIOP footprints in (e) and (f) are shown in (g), with the green lines highlighting the occurrence of smoke plumes. The crosses mark the location of Wuhan.

## Appendix B

In mid-October 2020, a series of smoke plumes formed from wildfires in California (Safford et al., 2022; Mamouri et al., 2023). The smoke layer appeared over Wuhan on 8-9 November originated from a thin smoke plume at altitudes of 12.5-14.0 km over the Mediterranean Sea on 27 October (red rectangle in Fig. A3(a)). Several plumes were also observed at altitudes of 5.0-14.0 km, as shown by UVAI data. The thin smoke plume in Fig. A3(a) can be further tracked back to the wildfire region in the northwest of America on 19 October, which is highly consistent with the HYSPLIT trajectories presented by Mamouri et al. (2023).

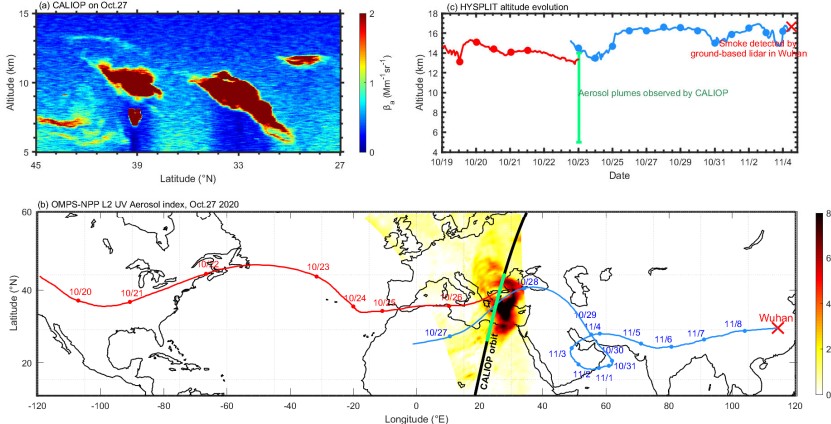

Figure B1. (a) CALIOP-observed 532-nm total attenuated backscatter coefficients on 27 October. (b) UVAI measured by OMPS on 27 October 2020. The 13-day backward trajectory starting from Wuhan at 16.7 km on 8 September and 8-day backward trajectory starting from 40.83°N, 26.36°E at 13.3 km on 27 October are presented. The CALIOP footprints in (a) are shown in (b), with the



green lines highlighting the occurrence of smoke plumes. The crosses mark the location of Wuhan.

## Data availability

CALIOP data can be obtained at the website https://search.earthdata.nasa.gov/search (CALIOP, 2024). OMPS data can be obtained at the website https://www.earthdata.nasa.gov/sensors/omps (OMPS, 2024). Copernicus Atmosphere Monitoring
Service (CAMS) reanalysis data can be obtained at the website https://www.ecmwf.int/en/forecasts/dataset/cams-global-reanalysis (CAMS, 2024). The HYSPLIT model is available at the website https://www.arl.noaa.gov (HYSPLIT, 2024). Lidar data used to generate the results of this paper are available from the authors upon request (e-mail: yf@whu.edu.cn).

## Author contributions

YH, DJ, and ZY analyzed the data and wrote the manuscript. ZY and KO participated in scientific discussions and reviewed
and proofread the manuscript. YH and FY conceived the research and acquired the research funding. FY leaded the study.

## Competing interests

The contact author has declared that none of the authors has any competing interests.

## Financial support

This work was supported by the National Natural Science Foundation of China (grant nos. 42005101, 41927804, and
42205130), the Chinese Scholarship Council (CSC) (grant no. 202206275006), the Hubei Provincial Natural Science Foundation of China (2023AFB617), and the Meridian Space Weather Monitoring Project (China).

## Acknowledgements

The authors thank the colleagues who participated in the operation of the lidar system at our site. We also acknowledge the Atmospheric Science Data Central (ASDC) at the NASA Langley Research Center for providing the CALIPSO data,
NASA/NOAA for the OMPS data, European Centre for Medium-Range Weather Forecasts (ECMWF) for Copernicus Atmosphere Monitoring Service ozone reanalysis data, and the NOAA Air Resources Laboratory (ARL) for the HYSPLIT model.

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
