# Peer review of "Long-term (2010-2021) lidar observations of stratospheric aerosols at Wuhan, China"

_EGUsphere, 2024_

## Author Comment (AC1)

**Response to reviewer #1**

**General Remarks**

*This manuscript describes a careful analysis of stratospheric aerosol observations and should be published after taking into consideration numerous comments listed below.*

**Response:** We appreciate the reviewer's thoughtful review and constructive comments. All the comments have been responded point by point as below and the corresponding modifications have been made in the revised manuscript.

**Specific Comments**

*Abstract: The first two sentences would better fit into the Introduction. You should restrict the text to presenting the key results.*

**Response:** We have removed the first three sentences and now start the abstract directly from the results for this study.

*Line 35: "Stratospheric aerosols play a critical role in the global radiation budget": I would add "during periods of strong loading". Otherwise, "critical" may mean an exaggeration.*

**Response:** "during periods of strong loading" has been added.

*Line 38: Please, add the paper of Jäger and Wege (J. Atmos. Chem., 10, 273–287, 1990).*

**Response:** This paper has been cited.

*Lines 47 to 49: The stratospheric loading strongly depends on the definition of the lower boundary chosen for the stratospheric aerosol. You should spend a few words on this issue. Trickl et al. (2013; 2024) determined a lower background after 2000 than other groups. This could point to a difference in algorithm in addition.*

**Response:** Thank you for pointing out this issue. We have added the following sentences **'Trickl et al. (2013; 2024) calculated the integrated backscatter coefficient (IBC) starting at 1 km above the tropopause since 2000, which is lower than the other groups. It is important to note that this differing definition of the lower boundary may result in a negative offset of IBC or sAOD.'** (please see lines 47-49)

*Line 60: "volcanic" (also lines 330 and 372)*

**Response:** "volcano" has been replaced by "volcanic".

*Line 63: Chouza et al and Trickl et al. report on lidar measurements only. I suggest to cite Kremser et al. here in order to include all the other techniques (or also Bingen et al., Remote Sensing of Environment 203 (2017) 296–321). Or add citations of papers on the individual techniques listed.*

**Response:** Thank you for your suggestions. Kremser et al. (2016) and Bingen et al. (2017) have been cited.

*Line 72: Lacking what? "still": Will this ever change? This remark is deceiving anyway because the reader might expect that the Wuhan system is going to close this gap (next sentence). Please, rephrase.*

**Response:** We have modified the statement as follows '**Since 2010, we have also conducted the long-term lidar observations of stratospheric aerosols in Wuhan (30.5°N, 114.4°E), central China, which can be a good supplementation to the geographical coverage of middle-latitude East Asian region.**' (please see lines 71-73)

*Line 94: "can be found in (Kong and Yi, 2015)"*

**Response:** "Kong and Yi, (2015)" has been modified to "(Kong and Yi, 2015)".

*Lines 93-94: A few technical data should be listed, maybe in a table. This would help the readers to make comparisons with other systems.*

**Response:** A table regarding the specifications of lidar has been added.

**Table 1. Specifications of the polarization lidar system at Wuhan University**

| Transmitter | | Receiver | |
|---|---|---|---|
| Laser | Continuum Inlite II-20 | Telescope | Cassegrain |
| Wavelength | 532 nm | Diameter | 300 mm |
| Energy/pulse | ~120 mJ | Field of view | 1 mrad |
| Repetition rate | 20 Hz | PMT | Hamamatsu 5783P |
| Pulse duration | 6 ns | Digitizer | Licel TR40-160 |

**Reference:**

Kong, W. and Yi, F.: Convective boundary layer evolution from lidar backscatter and its relationship with surface aerosol concentration at a location of a central China megacity, J. Geophys. Res. Atmos. 120, 7928–7940, 2015.

He, Y., and Yi, F.: Dust aerosols detected using a ground-based polarization lidar and CALIPSO over Wuhan (30.5°N, 114.4°E), China. Adv. Meteorol., 2015, 536762, 2015.

*Line 95: 24/7 is not self-explaining!*

**Response:** '24/7 routine observations' have been revised to 'continuously'.

*Line 100-102: To determine the tropopause from aerosol measurements is in conflict with its definition. Not always an aerosol edge is seen at the tropopause.*

**Response:** For clarity, we have revised this sentence as follows '**Same as Trickl et al. (2024), we use 1 km above tropopause as the lower limit for sAOD calculation to avoid the possible influence of tropospheric aerosols and to incorporate the stratospheric aerosols as much as possible.**

**Therefore, the stratospheric aerosol optical depth (sAOD) is calculated by integrating the aerosol extinction coefficient from 1 km above tropopause to 30 km to minimize disturbances from the troposphere and ensure a sufficient signal-to-noise ratio (SNR).**' (please see lines 103-107)

*Line 115: "lidar ratio of 50 sr (non-smoke) or 70 sr (smoke)": This was already mentioned above! I think that it makes sense to write this exclusively here.*

**Response:** "with a lidar ratio of 50 sr (non-smoke) or 70 sr (smoke)" has been removed.

*Line 122: Please, give a few details of the ozone data. Is there a seasonal cycle? Etc.*

**Response:** For clarity, we have added the following sentences regarding ozone data '**We utilized $n_{O_3}$ profiles from the Copernicus Atmosphere Monitoring Service (CAMS) reanalysis data within the latitudes of 28.0-31.8°N and longitudes of 113.0-114.5°E. To correct aerosol extinction by considering ozone absorption, the seasonal average $n_{O_3}$ profiles from 2015 were employed as the representative ozone levels for each season through our measurement period, i.e., March-April-May for spring, June-July-August for summer, September-October-November for autumn, and December-January-February for winter. The relative uncertainty of the backscatter coefficient profile was less than 5%.**' (please see lines 123-128)

*Line 126: "number density"!*

**Response:** 'number concentration' has been modified to 'number density'.

*Figure 2: Does the profile really end at 22 km? In other publications you find much higher upper aerosol boundaries.*

**Response:** Yes, the profile ends at 22 km. This figure only shows a typical result regarding the correction of aerosol extinction by considering ozone absorption. Only during intensive stratospheric intrusion events, higher upper aerosol boundaries can be found.

*Line 144: A reference is missing.*

**Response:** A reference has been added.

**Reference:**

Jaross, G., Bhartia, P. K., Chen, G., Kowitt, M., Haken, M., Chen, Z., Xu P., Warner, J., and Kelly, T.: OMPS Limb Profiler instrument performance assessment. J. Geophys. Res. Atmos., 119(7), 4399-4412. https://doi.org/10.1002/2013JD020482, 2014.

*Line 157: The HYSPLIT team ask to cite also a more recent publication: Stein, A. F., Draxler, R. R, Rolph, G. D., Stunder, B. J. B., Cohen, M. D., and Ngan, F.: NOAA's HYSPLIT atmospheric transport and dispersion modeling system, B. Am. Meteorol. Soc., 96, 2059– 2077, 2015.*

**Response:** Thank you for providing the reference. 'Stein et al. (2015)' has been cited.

*Line 165: Why not radiosonde data? Please, explain. Is the tropopause used in the analysis? I see 17 km as the start altitude (line 103).*

**Response:** According to the current regulation, the radiosonde data in China above 17 km latitude cannot be publicly provided (especially from 2020 onwards), as seen from the data downloaded from the website of the University of Wyoming. This is the reason we show the tropopause altitude (in Fig. 3) from the CALIOP (before August 2020) and OMPS (after August 2020) Level-3 data product (provided by GMAO MERRA-2 reanalysis). Showing the tropopause altitudes is to verify the justification of taking 17 km as the lower boundary in sAOD calculation.

*Line 167: "persistently appeared during that period" would clarify that the sentence is about the Wuhan result.*

**Response:** 'persistently appeared during that period' has been added.

*Line 168: "with more abundant", i.e. relative to the background.*

**Response:** "more" has been added.

Line 172: I suggest to add the start altitude for the determination of the sAOD also here.

**Response:** We have added the start altitude for determining the sAOD.

*Line 173: Please, add "(Newhall and Self, 1982)": The volcanic explosivity index (VEI): an estimate of explosive magnitude for historical volcanism, J. Geophys. Res., 87, 1231-38, 1982.*

**Response:** Newhall and Self (1982) has been cited.

*Line 175: "volcanoes" (also line 204)*

**Response:** "volcanos" has been modified to "volcanoes".

*Lines 179-180: "cannot be attributed to a significant influence of Kelud"*

**Response:** "the influence of Kelud" has been modified to "a significant influence of Kelud".

*Line 181: "but" must be preceded by a comma if there is a second verb.*

**Response:** A comma was added before "but".

*Line 193-194: **"The period between January 2013 and August 2017 represents the stratospheric-quiescent period."** Better omit or explain in the text since this period is not specified in the table. The data evaluation (Rayleigh calibration) at low latitudes (Mauna Loa) must be made up to 40 km and more.*

**Response:** For clarity, we have revised the last sentence in the caption '**In Wuhan, the period from 2010 to 2021 represents the entire lidar measurement period, while the period from January 2013 to August 2017 represents a local (only for Wuhan) stratospheric-quiescent period.**' (please see lines 199-202)

As for Rayleigh calibration in Mouna Loa, Chouza et al., (2020) took a reference altitude range of 35-37 km for Raman retrieval and 38-40 km for Klett retrieval. They calculated sAOD by integrating the extinction coefficient from 17 to 33 km. The details can be found in sections 3.1 and 3.2 therein. Here we give 17-33 km to show how sAOD is calculated for Mouna Loa results.

*Table 2: Please, explain the upper end of 25 km for Wuhan better in the text (line 199). For Garmisch-Partenkirchen no period is given in the table, just the 1979 background. The GP data are available on the NDACC web site until the end of 2023 (Trickl et al., 2024). The lidar was based on a ruby laser until 1990 and on 532 nm later on. Just for the integrated backscatter coefficient (IBC) a conversion from 532 nm to the ruby wavelength was made. How did you make this conversion (the IBCs are not published)? Please, remove the H2O DIAL: it was just used for a case study.*

**Response:** We are grateful for your valuable comments. The former aerosol properties were obtained by meteorological data from the Global Data Assimilation System (GDAS1), which was up to 25 km. We have updated the aerosol properties to 30 km, by connecting U.S. Standard Atmosphere (1976) above 25 km. We have modified the related sentence as below '**The mean background sAOD (1 km above tropopause to 30 km) over Wuhan was 0.0044 (±0.0019), as obtained from January 2013 to August 2017. '** (please see lines 205-206)

For stratospheric aerosol observations, usually, only the Mie channel is available for different lidar systems at different sites. Thus, we have removed the third column of Table 3 regarding 'instrument'. As for sAOD over Garmisch-Partenkirchen, the long-term variation of IBC values is given in the figure and we do not find a published average value of IBC or sAOD. Therefore, we only provide 694-nm sAOD in 1979 calculated from IBC by multiplying by a lidar ratio of 50 sr as a reference value of background stratospheric aerosol level. Accordingly, we have modified Table 3 as well as the caption.

Moreover, the $H_2O$ DIAL lidar has been removed according to the reviewer's suggestion.

*Line 200: "the AOD"*

**Response:** "AOD" has been modified to "the AOD".

*Lines 204-205: The examples given are not in the tropics. Better split into two sentences. It is interesting that no higher -latitude eruption occurred before 2006 which led to very low backscatter coefficients at northern mid-latitudes after the end of the Pinatubo period. Quite obviously, the tropical contributions were not significantly transported northward.*

**Response:** The related statements have been revised as follows '**It is interesting to note that before 2006, volcanic eruptions with VEI ≥4 mainly occurred in the tropics (Chouza et al., 2020), and**

**did not cause a noticeable enhancement of sAOD (or IBC) at mid-latitude sites in the Northern Hemisphere, revealing that topical volcanic aerosols emitted during these events were not significantly transported northward.'** (please see lines 210-212)

*Line 206: Add "above Wuhan".*
**Response:** "above Wuhan" has been added.

*Line 216: Please, explain ATAL.*
**Response:** We have defined the abbreviation 'ATAL' as 'Asian tropopause aerosol layer' for the first presence in the introduction section.

*Line 225: "The Nabro volcano"*
**Response:** "Nabro volcano" has been modified to "The Nabro volcano"

*Line 232: What does the "proportional to" sign preceding "130" mean? Please, rephrase.*
**Response:** '~' has been replaced by 'approximately'.

*Line 241: Ohneiser et al. (Atmos. Chem. Phys., 21, 15783–15808, 2021) discuss the presence of biomass-burning aerosol during that period. Is this part of the "main aerosol plume". Please, explain this phrase and address the findings of Ohneiser et al.*
**Response:** We appreciate your constructive comments. The answer is no. According to the results from the literature, we have compared the optical properties of Raikoke volcanic aerosols and Siberian wildfire smoke in 2019 as shown in Jing et al. (2023) (see Table 1 therein). In general, Siberian wildfire smoke showed much larger AODs and lidar ratio than Raikoke volcanic aerosols (Ansmann et al., 2021; Ohneiser et al., 2021; Vaughan et al., 2021; Kloss et al., 2021). In addition, Ohneiser et al. (2021) discussed that large parts of the smoke were transported into the central Arctic and were trapped by the polar vortex. Therefore, less smoke was transported to the low latitudes. For clarity, the following sentences have been added '**Note that intense Siberia wildfire took place meanwhile in the summer of 2019 (19 July to 14 August). Ohneiser et al. (2021) and Ansmann et al. (2024) found that large parts of the smoke were transported into the central Arctic and were trapped by the polar vortex; thus, less smoke was transported to the low latitudes. Moreover, Jing et al. (2023) have discussed that the stratospheric aerosol plumes observed over Wuhan are probably only from the Raikoke eruption. Because the plume-isolated 532-nm AODs for Siberian smoke are approximately 0.1 as observed in Leipzig (Ansmann et al., 2021) and in the Arctic (Ohneiser et al., 2021), which are much larger than those for Raikoke volcanic aerosol layers observed in Wuhan (0.001-0.017, Jing et al., 2023), Leipzig (0.010-0.015, Ansmann et al., 2021), and Capel Dewi Atmospheric Observatory in UK (0.01-0.05, Vaughan et al., 2021).'** (please see lines 251-258)

**References:**

Ansmann, A., Ohneiser, K., Chudnovsky, A., Baars, H., and Engelmann, R.: CALIPSO aerosol-typing scheme misclassified stratospheric fire smoke: case study from the 2019 Siberian wildfire season. Front. Environ. Sci. 9, 769852 https://doi.org/10.3389/fenvs.2021.769852, 2021.

Ansmann, A., Veselovskii, I., Ohneiser, K., and Chudnovsky, A.: Comment on "stratospheric aerosol composition observed by the atmospheric chemistry experiment following the 2019 Raikoke eruption" by Boone et al. J. Geophys. Res. Atmos., 129, e2022JD038080. https://doi.org/10.1029/2022JD038080, 2024.

Jing, D., He, Y., Yin, Z., Liu, F., Yi, Y., and Yi, F.: Evolution of aerosol plumes from 2019 Raikoke volcanic eruption observed with polarization lidar over central China, Atmos. Environ., 119880. https://doi.org/10.1016/j.atmosenv.2023.119880, 2023.

Ohneiser, K., Ansmann, A., Chudnovsky, A., Engelmann, R., Ritter, C., Veselovskii, I., Baars, H., Gebauer, H., Griesche, H., Radenz, M., Hofer, J., Althausen, D., Dahlke, S., and Maturilli, M.: The unexpected smoke layer in the High Arctic winter stratosphere during MOSiC 2019-2020. Atmos. Chem. Phys. 21, 15783–15808. https://doi.org/10.5194/acp-21-15783-2021, 2021.

Vaughan, G., Wareing, D., and Ricketts, H.: Measurement Report: Lidar measurements of stratospheric aerosol following the 2019 Raikoke and Ulawun volcanic eruptions, Atmos. Chem. Phys. 21, 5597–5604. https://doi.org/10.5194/acp-21-5597-2021, 2021.

*Line 251: "self-lofting" is not a good expression although it has been used in the literature. Please, rephrase, explain and add one or more references.*

**Response:** Thank you for pointing this out. We cannot fully confirm the reason for the higher altitudes of the second arrival of CCC compared to the first. Therefore, we have removed the expression 'self-lofting'.

*Line 264 (and 286): I would write "represents" instead of "denotes". "To denote" is usually used in connection with language.*

**Response:** 'denote' has been replaced by 'represent'.

*Line 319: "Here,"*

**Response:** 'Here' has been modified to 'Here,'.

*Lines 322-323: The Junge layer was already introduced before with similar words. Therefore, you can write something like "This non –seasonal background is interpreted as the Junge layer."*

**Response:** It has been written as '**This non-seasonal background is interpreted as the Junge layer, a global-wide stratospheric aerosol layer at around 20 km altitude.**' (please see lines 335-336)

Line 355: Replace "which" by "that".

**Response:** 'which' has been replaced by 'that'.

*Lines 372-374: "before 2010 statistics by Chouza et al. (2020)": Something is missing here.*
*"While": Do you mean "By contrast"?*

**Response:** 'before 2010 statistics by Chouza et al. (2020)' has been modified to 'before 2010 as reviewed by Chouza et al. (2020)'. 'While' has been replaced by 'By contrast'.

*Line 382: "the Observatoire". "the" can be omitted if you use OHP instead.*
**Response:** "Observatoire de Haute-Provence" has been modified to "OHP".

*Line 384: "were estimated".*
**Response:** "was" has been modified to "were".

*Line 387: "volcanoes"*
**Response:** "volcano" has been modified to "volcanoes".

Line 416: "the build-up of the ATAL"?
**Response:** "the ATAL" has been modified to "the build-up of the ATAL".

Line 421: "the polarization lidar"
**Response:** "the" has been added.

Line 428: "extinction coefficients and lidar ratios"
**Response:** "extinction and lidar ratio" has been modified to "extinction coefficients and lidar ratios"

Line 429: "a larger telescope"
**Response:** "a" has been added.

Lines 429-430: I hope that this promise is underlined by the reference given (or other work). It terribly hard to determine stratospheric aerosol extinction coefficients from Raman scattering. This was demonstrated by the Geesthacht group during the very special Pinatubo period. Why do you not consider to install an HSR channel? You could suggest this here in addition.

**Response:** We are grateful for your valuable comment. High Spectral Resolution Lidar (HSRL) is a powerful instrument for accurate profiling of aerosol extinction coefficient; however, it would require more investment in the upgrade of transmitting and receiving modules of the lidar system. We have focused on rotational Raman lidar techniques for temperature profiling since 2010, which cost most of our budget, and we do not have additional budgets for the upgrade. As an outlook, we have modified the related sentence to '**In addition, more accurate aerosol extinction coefficients and lidar ratios can be obtained with a high spectral resolution lidar.**' (please see lines 446-447)

---

## Author Comment (AC2)

**Response to reviewer #2**

**General Remarks**

*This paper describes 11 years of lidar data at 30N in eastern China and compares the measurements to several satellite and other lidar records. The period includes several volcano and wild fire events.*

*The primary difficulty with the paper is in the variable altitude intervals used in the sAOD comparisons. First it is not clear why the Wuhan data above 25 km are not used for the sAOD calculation, while all other sites extend their sAOD calculations to 30 km or above. Second there is often a lot of stratosphere and aerosol below 17 km where the Wuhan calculations begin, particularly in the winter months, yet this region also seems to be ignored. Why is that when other records extend to the tropopause or 1 km above the tropopause? The impact of ignoring these differences on the sAOD comparisons is not even mentioned, yet it may contribute significantly to the differences which are observed.*

**Response:** We appreciate the reviewer's thoughtful review and constructive comments. We use the meteorological data from the Global Data Assimilation System (GDAS1) for aerosol retrieval, which can only reach up to 25 km altitude. Now, by using U.S. Standard Atmosphere (1976) above 25 km, we have updated the retrieved aerosol properties to 30 km, which is considered the upper limit for sAOD calculation.

In addition, we have used tropopause+1 km as the lower limit for sAOD calculation in the revised manuscript. As a result, the sAOD values integrated from tropopause+1 km to 30 km have been updated in Fig. 3 and we have also added the statements below '**Same as Trickl et al. (2024), we use 1 km above tropopause as the lower limit for sAOD calculation to avoid the influence of tropospheric aerosols and to incorporate the stratospheric aerosols as much as possible. Therefore, the stratospheric aerosol optical depth (sAOD) is calculated by integrating the aerosol extinction coefficient from 1 km above tropopause to 30 km to minimize disturbances from the troposphere and ensure a sufficient signal-to-noise ratio (SNR).**' (please see lines 103-107).

[Figure]

Figure 3. (a) Time-height contour plots of the aerosol backscatter coefficient measured by 532-nm polarization lidar over Wuhan during 2010-2021; the white curve represents the monthly mean tropopause from CALIOP (October 2010 to July 2020) and OMPS (August 2020 to September 2021). (b) The evolution of monthly mean 532-nm sAOD from 1 km above tropopause to 30 km derived from polarization lidar observation (black curve) at Wuhan. The red dashed line represents the background sAOD of 0.0038.

**Specific Comments**

*Here are additional detailed comments to address.*

*Figure 1. The labels on some lidar sites may be misleading. Are both the Hampton and Sao Jose dos Campos sites still making measurements? If not then indicate the time frame of measurement availability.*

**Response:** The observation periods at Hampton and Sao Jose dos Campos sites have been modified. Sakai et al. (2016) and Hofer et al. (2024) mentioned that the lidar system at Lauder station is working continuously; thus, we mark it as 'since 1992' in Fig. 1.

[Figure]

Figure 1. The locations of ground-based lidar sites with long-term stratospheric aerosol observations (solid dots) and two main volcanic eruptions, i.e., Nabro 2011 and Raikoke 2019 (solid triangles), as reported by Kremser et al. (2016), Hofer et al. (2024), and Trickl et al. (2024).

**Reference:**

Hofer, J., Seifert, P., Liley, J. B., Radenz, M., Uchino, O., Morino, I., Sakai, T., Nagai, T., and Ansmann, A.: Aerosol-related effects on the occurrence of heterogeneous ice formation over Lauder, New Zealand∕Aotearoa, Atmos. Chem. Phys., 24, 1265–1280, https://doi.org/10.5194/acp-24-1265-2024, 2024.

Kremser, S., Thomason, L. W., von Hobe, M., Hermann, M., Deshler, T., Timmreck, C., Toohey, M., Stenke, A., Schwarz, J. P., Weigel, R., Fueglistaler, S., Prata, F. J., Vernier, J., Schlager, H., Barnes, J. E., Antuña-Marrero, J., Fairlie, D., Palm, M., Mahieu, E., Notholt, J., Rex, M., Bingen, C., Vanhellemont, F., Bourassa,

A., Plane, J. M. C., Klocke, D., Carn, S. A., Clarisse, L., Trickl, T., Neely, R., James, A. D., Rieger, L., Wilson, J. C., and Meland, B.: Stratospheric aerosol—Observations, processes, and impact on climate, Rev. Geophys., 54, 278–335. https://doi.org/10.1002/2015RG000511, 2016.

Trickl, T., Vogelmann, H., Fromm, M. D., Jäger, H., Perfahl, M., and Steinbrecht, W.: Measurement report: Violent biomass burning and volcanic eruptions – a new period of elevated stratospheric aerosol over central Europe (2017 to 2023) in a long series of observations, Atmos. Chem. Phys., 24(3), 1997–2021. https://doi.org/10.5194/acp-24-1997-2024, 2024.

*166-167. The explanation for the source of the sulfur for the stratospheric aerosol layer isn't correct. While tropospheric SO2 plays a, still unquantifiable role, H2S does not get to the stratosphere. The primary source of stratospheric sulfur is OCS. See any of the review papers on stratospheric aerosol e.g. Kremser et al., 2016 or the SPARC Assessment of Stratospheric Aerosol Properties (Thomason and Peter).*

**Response:** The "$H_2S$" has been modified to "OCS (carbonyl sulfide)".

*Fig. 3 and discussion on sAOD. The authors need to discuss and perhaps quantify the fraction of sAOD ignored during the winter by limiting their integration to 17-25 km. In the winter there is up to 4 to 5 km of the atmosphere ignored in this formulation as the tropopause descends in winter.*

**Response:** Considering the reviewer's comments, we have recalculated the sAOD by setting a lower limit of troposphere+1km. Accordingly, the sAOD values have been updated in the revised manuscript.

*Table 3 suffers from similar problems. All the mid latitude lidars save Haute Provence calculate AOD by integrating from near the tropopause to > 30 km. The Mauna Loa lidar is integrated from 17 km, but this is a tropical site and the tropopause varies little from 17 km throughout the year. Without having similar altitude integral it is unclear what the value is in comparing sAODs. Also, the sAODs are not just one number. What is reported, the mean, median, is there a standard deviation, ...?*

**Response:** We have changed the lower limit of sAOD calculation from 17 km altitude to 1 km above the tropopause. The former aerosol retrievals were based on the meteorological data from the Global Data Assimilation System (GDAS1), which can only reach up to 25 km altitude. We have updated the aerosol retrievals to 30 km, by using the U.S. Standard Atmosphere (1976) above 25 km altitude. Therefore, we have recalculated the sAOD by integrating the aerosol extinction coefficient from 1 km above the tropopause to 30 km.

We have added/modified the related statements as follows "**The mean background sAOD (1 km above tropopause to 30 km) over Wuhan was 0.0044 (±0.0019), as obtained from January 2013 to August 2017.** " (please see lines 204-206).

*Fig. 9 Is there a mistake in the abscissa label? Should it be Mm$^{-1}$sr$^{-1}$ as in all the other plots? What does the shading represent? If the standard deviation then wouldn't it be much larger? Notice in the previous plots the backscatter coefficient exceeds values of 1 Mm$^{-1}$sr$^{-1}$ in a number of cases.*

**Response:** Thank you very much for pointing out the mistake. The unit of the abscissa label in Fig. 9 has been modified to 'Mm$^{-1}$sr$^{-1}$'. The shadings have been modified to show the standard deviations; we have added the related statements in the caption of Fig. 9. Although the backscatter coefficient of CCC in the Raikoke event (2019) exceeded 1 Mm$^{-1}$sr$^{-1}$, it was only an occasional case with a relatively short duration compared to the entire period generating the statistical results.

*Fig. 10 The altitude interval over which these calculations were made should be mentioned. The quantities in the box and whisker plots should be defined in the caption. What is the box, the center line, ...?*

**Response:** The caption of Fig. 10 has been modified to "**Monthly mean sAOD integrated from 1 km above tropopause to 30 km in the cold half-year (October-next March) and warm half-year (April-September). For each box, the center lines represent the median value, and the bottom and top edges of the box represent the 25th and 75th percentiles, respectively. The whiskers were set to be 1.5.**"

*359 ...60 W/m$^2$ for BC ...*

**Response:** The "and" has been modified to "for".

*363 How are sAOD_OC and sAOD_BC determined from the lidar data?*

**Response:** The contribution of OC or BC cannot be directly distinguished by polarization lidar without the assumption. The ratio of OC and BC and the conversion factor were provided by Koch et al. (2001) and Hanson et al., (2005), the related statements have been given in section 3.5 "**As estimated by Hanson et al. (2005), the conversion factors from AOD to RF are -13 W·m$^{-2}$ for OC and 60 W·m$^{-2}$ for BC. As a result, the contribution of sAOD to RF can be divided into three parts: background sAOD (sAOD$_{background}$), OC sAOD (sAOD$_{OC}$) and BC sAOD (sAOD$_{BC}$). The RF during the smoke injection period can be calculated as follows:**

$$\mathbf{RF_{smoke} = sAOD_{background} \times (-25) + sAOD_{OC} \times (-13) + sAOD_{BC} \times 60}$$

(please see lines 376-380).